 **eLIFE**

# Human motor fatigability as evoked by repetitive movements results from a gradual breakdown of surround inhibition

**Marc Bächinger[1,2†*], Rea Lehner[1,2†*], Felix Thomas[1,2], Samira Hanimann[1], Joshua Balsters[1,3], Nicole Wenderoth[1,2*]**

[1]Department of Health Sciences and Technology, Neural Control of Movement Lab, Zurich, Switzerland; [2]Neuroscience Center Zurich (ZNZ), University of Zurich, Federal Institute of Technology Zurich, University and Balgrist Hospital Zurich, Zurich, Switzerland; [3]Department of Psychology, Royal Holloway University of London, Egham, United Kingdom

**Abstract** Motor fatigability emerges when demanding tasks are executed over an extended period of time. Here, we used repetitive low-force movements that cause a gradual reduction in movement speed (or 'motor slowing') to study the central component of fatigability in healthy adults. We show that motor slowing is associated with a gradual increase of net excitability in the motor network and, specifically, in primary motor cortex (M1), which results from overall disinhibition. Importantly, we link performance decrements to a breakdown of surround inhibition in M1, which is associated with high coactivation of antagonistic muscle groups. This is consistent with the model that a loss of inhibitory control might broaden the tuning of population vectors such that movement patterns become more variable, ill-timed and effortful. We propose that the release of inhibition in M1 is an important mechanism underpinning motor fatigability and, potentially, also pathological fatigue as frequently observed in patients with brain disorders.
DOI: https://doi.org/10.7554/eLife.46750.001

**\*For correspondence:**
marc.baechinger@hest.ethz.ch (MBä);
rea.lehner@hest.ethz.ch (RL);
nicole.wenderoth@hest.ethz.ch (NW)

†These authors contributed equally to this work

**Competing interests:** The authors declare that no competing interests exist.

## Introduction

Motor fatigability is a phenomenon experienced in everyday life when exhaustive exercise or physically demanding tasks have to be maintained for extended time periods. Enhanced motor fatigability is a prevalent symptom of many brain disorders (such as stroke, Parkinson's disease, or traumatic brain injury), which typically affects submaximal movements that are required for many daily life tasks (*Kluger et al., 2013*; *Manjaly et al., 2019*). In addition to clinical data, experimental evidence in healthy participants has shown that fatigability arises, at least partly, at the supraspinal level suggesting that the descending drive from motor cortex is suboptimal once it is fatigued (*Gandevia et al., 1996*; *Smith et al., 2007*; *Søgaard et al., 2006*). While this reduction in central drive has been associated with diverse activity changes within cortico-subcortical networks (*van Duinen et al., 2007*; *Post et al., 2009*), our current understanding of the neurophysiological mechanisms underlying central fatigability is still limited.

Here, we investigate the neurophysiological mechanisms associated with performance fatigability of repetitive submaximal movements (i.e. the objectively measurable performance decrease which manifests when demanding tasks are maintained over an extended period of time; *Kluger et al., 2013*). It has been demonstrated previously that the performance of repetitive movements tends to deteriorate over time (*Dolan and Adams, 1998*; *Miller et al., 1993*). While this finding is not unexpected for fatiguing contractions (e.g., at high force levels), a similar phenomenon has been demonstrated for movements executed with submaximal forces but at high tapping speed. For example,

7–9 s of finger tapping at the maximal voluntary rate is sufficient to induce a significant performance decrease (*Aoki et al., 2003*; *Rodrigues et al., 2009*). A similar phenomenon also emerges for skilled motor tasks such as motor sequence tapping involving multiple fingers, a task where the tapping rate of each finger is well below the maximal frequency observed for single digit tapping (*Brawn et al., 2010*). Once the finger sequence is over-learned, the initial tapping speed increases but a pronounced pattern of slowing is observed during a period of tapping for 30 s. We will refer to this characteristic decrease in movement speed as 'motor slowing' and use the term (performance) fatigability to refer to more general mechanisms of reversible forms of fatigue.

The neurobiological underpinnings of motor slowing are largely unknown. Previous studies have shown that motor slowing is robustly evoked by prolonged finger tapping at high speed but markers of peripheral or muscular fatigability are virtually unchanged (*Arias et al., 2015*; *Madrid et al., 2016*; *Rodrigues et al., 2009*), giving rise to the hypothesis that supraspinal mechanisms play a major role in evoking this phenomenon. Moreover, slowed motor execution is a hallmark of healthy aging (*Mattay et al., 2002*; *Yordanova et al., 2004*) and associated with a dysregulation of motor cortex excitability (*Teo et al., 2012a*). Together, these findings point towards a supraspinal locus of the phenomenon but it is still unclear which neurophysiological or computational mechanisms cause motor slowing during repetitive movements. Here we aim to unravel the neurobiological underpinnings of motor slowing using a multimodal approach involving functional magnetic resonance imaging (fMRI) to identify which whole-brain networks might mediate motor slowing, electroencephalography (EEG) to measure cortical activity during recovery from motor slowing and, finally, transcranial magnetic stimulation (TMS) to probe changes within different cortical circuits in primary motor cortex (M1). We show that motor slowing is a general phenomenon that gradually manifests when high tapping speeds have to be maintained for an extended period of time. It is observed independent of the effectors or muscle groups involved, and also independent of the complexity of the repetitive movement task. Further we show in a series of functional imaging and neurophysiological experiments that substantial motor slowing is associated with an increase of the excitation-inhibition ratio within the motor network and, particularly, in M1. Our main finding is that motor slowing is accompanied by a breakdown of surround inhibition in M1 and an increase in coactivation of antagonistic muscle groups. We suggest that this breakdown in motor control followed by an increase in cocontraction of antagonistic muscle groups is tightly associated with the slowing of repetitive movements.

## Results

First, we characterised the time and speed dependency of motor slowing. In experiment 1 (n = 23) participants were asked to tap repetitively with their index and middle finger for 10 s, 30 s or 50 s at their maximal speed followed by a 50 s break. Calculating movement speed within 1 s bins revealed a characteristic speed reduction (slowing) which manifests immediately after movement onset and reaches a 'plateau' after approximately 30 s (*Figure 1A*), demonstrating that motor slowing is a time-dependent phenomenon which is only mildly expressed after the first 10 s while the full slowing effect is observed after 30 s. Next, we tested the influence of the initial tapping speed on the motor slowing phenomenon (experiment 2, N = 12). Participants tapped with their index finger for 30 s starting at their maximal tapping speed which revealed a strong slowing effect in all participants (*Figure 1B*, blue). Next, the same participants performed 30 s tapping trials while receiving online feedback of their tapping speed together with a pace representing either their initial tapping speed (Fast pacing, *Figure 1B* green), their final speed (Slow pacing, *Figure 1B* pink) or 90% of their end speed (Ultraslow pacing, *Figure 1B* red). Importantly, a pronounced expression of slowing is only observed for high tapping speeds while paced tapping at lower speeds causes virtually no slowing (*Figure 1B*, significant *condition x time* interaction (F(6,121)=49.314, p<0.001), *supplementary file 1*). Together experiment 1 and 2 indicate that motor slowing is a gradual process which depends inherently on the initial speed and the time it is maintained. Thus, motor slowing exhibits two hallmark features of performance fatigability as demonstrated previously for tasks impacting strongly on peripheral neuromuscular factors, for example, sustained isometric contractions at high force levels (*Enoka and Stuart, 1992*).

Based on these initial results, all further experiments (*Figure 2A*) required healthy young volunteers to perform repetitive movements at maximal speed either for a period of at least 30 s evoking

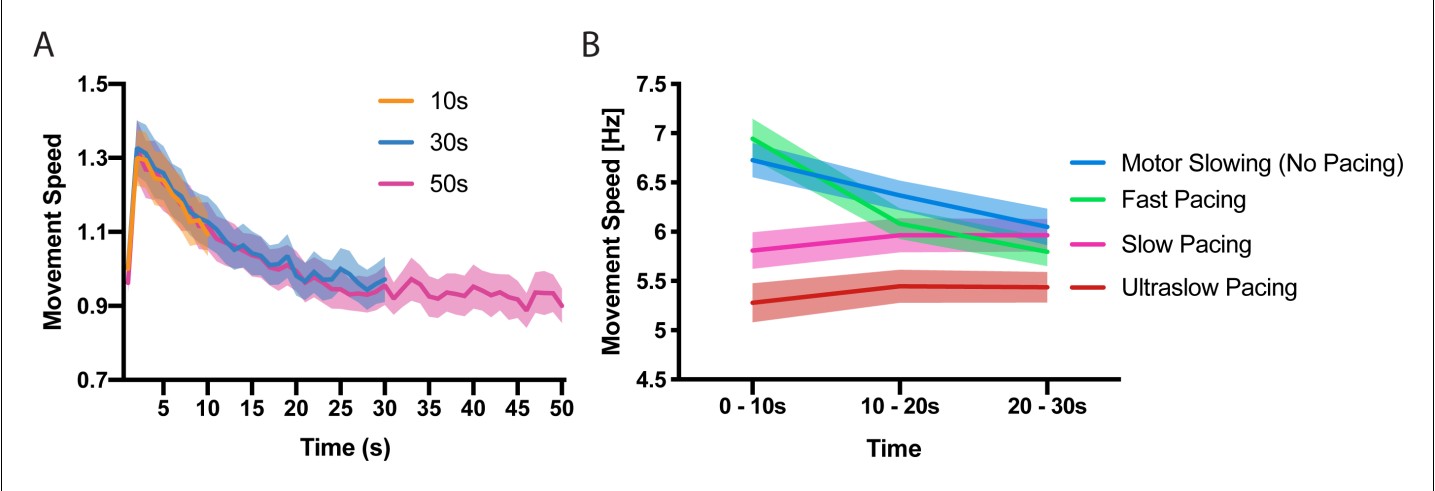

**Figure 1.** Time and speed dependency of motor slowing. (A) Time course of motor slowing. Participants tapped at maximum speed for 10 s (orange), 30 s (blue) or 50 s (pink) at their maximum speed. Slowing starts immediately with movement onset and reaches a 'plateau' after about 30 s. All data normalised to first second of 10 s condition. (B) Motor slowing depends on initial movement speed. After performing motor slowing trials starting at maximum speed with no online feedback (Motor Slowing, that is no pacing, blue), participants were paced at their initial speed (Fast pacing, green), their end speed (slow pacing, pink) or 90% of their end speed (Ultraslow pacing, red). Note that participants adjust to the pacing speed during the first time bin (0–10 s) and typically underpace for the slow and ultraslow pacing conditions while they tend to overpace for the fast pacing condition (see also *Figure 1—figure supplement 1* for an additional analysis on participants compliance with the pacing frequency and *supplementary file 1* for individual participant data). All values mean ± sem.

DOI: https://doi.org/10.7554/eLife.46750.002

The following figure supplement is available for figure 1:

**Figure supplement 1.** Deviation from the target speed in the pacing experiment shown in *Figure 1B*.

DOI: https://doi.org/10.7554/eLife.46750.003

substantial slowing (slowing condition) or for only 10 s evoking only minor slowing (control condition). Both conditions were followed by a 30 s break (recovery period). The presence of motor slowing was defined as a significant reduction of movement speed (measured in movement cycles per 10 s; see Materials and methods) over the course of the slowing conditions in comparison to the much shorter control conditions. Note that a minor reduction of movement speed is still present during the control condition, albeit significantly less pronounced than in the actual slowing condition.

We then asked whether motor slowing is a general phenomenon that can be observed irrespective of the effector performing the repetitive movement at maximal speed. We performed three experiments involving three different effectors (experiments 3–5). In the first experiment, participants (n = 12) executed repetitive alternating left and right foot taps (Figure B, experiment 3). In the second experiment, participants performed leftward and rightward saccades (*Figure 2C*, experiment 4), and in the third experiment they performed alternating index and middle finger taps (*Figure 2D*, experiment 5). All tasks caused significant motor slowing of about 20% over a period of 30 s (foot: 13.9 ± 2.7% Cohen's d = 1.88; eyes: 18.3 ± 2.8%, Cohen's d = 3.35; finger: 21.6 ± 6.4% Cohen's d = 2.48, all values mean ± sem; linear mixed-effects model (LMEM) summarised results for experiments 1–3; $F_{(2,11)} \geq 21.813$, p<=0.001, *Figure 2B–D*.).

Interestingly, we observed virtually no accumulation of motor slowing over the course of the experiment, that is the initial tapping speed measured during the first 10 s of each trial was largely unchanged (see *Table 1*; *time effect*: p≥0.134). This suggests that the process causing motor slowing is able to spontaneously recover during the subsequent 30 s break. Importantly, in experiment five we also modulated the break length (i.e. 5 s, 10 s, 15 s, 20 s, 25 s, 30 s break lengths, randomised within participants; see Materials and method section for details) after both the slowing and the control conditions (experiment 5), which allowed us to further investigate the time course of recovery. We found that the break length significantly influenced subsequent tapping speed (*break*

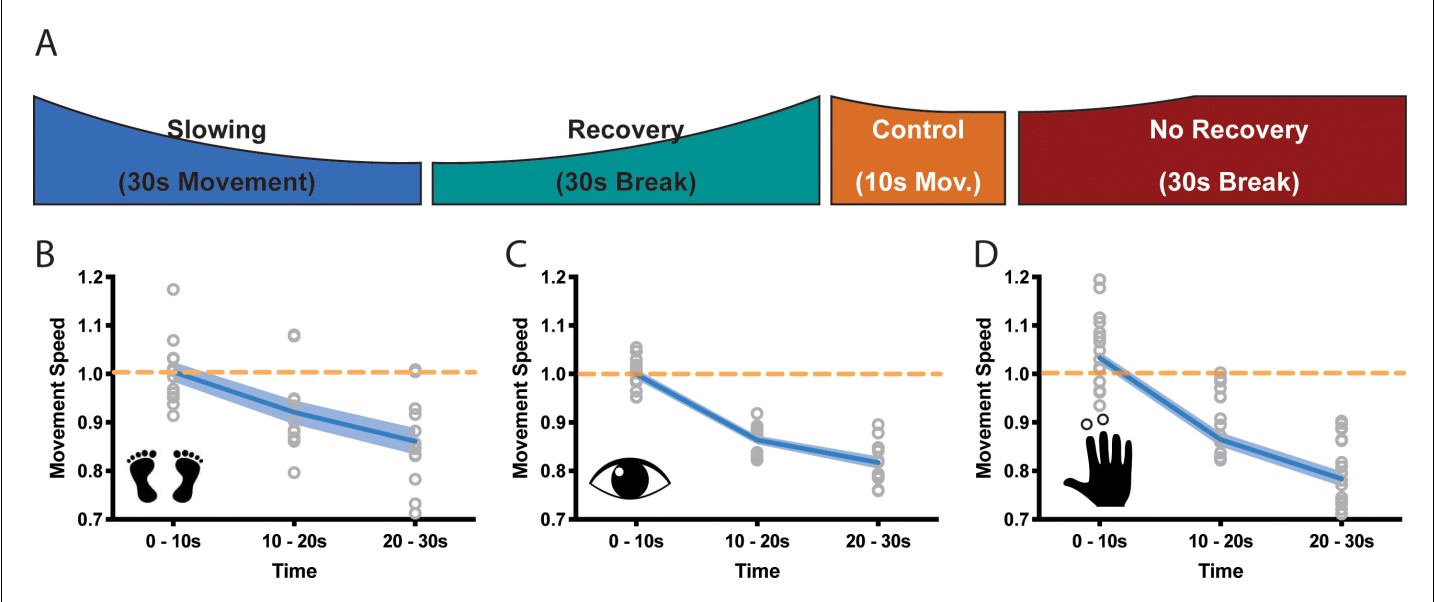

**Figure 2.** Motor slowing of different movement effectors. (**A**) General behavioural paradigm. Participants were asked to either perform long (≥30 s) or short (10 s) blocks of repetitive movements, followed by breaks of at least 30 s. Movement speed was analysed in movement cycles per 10 s. (**B–C**) Motor Slowing (blue line) normalised to control condition (orange line) occurred independent of the involved effector and was present during foot tapping (B, N = 12), repetitive eye movements (C, N = 12) and repetitive finger movements (D, N = 17). Note that the data in (**D**) corresponds to the experiment depicted in **Figure 3**, where subjects had a break of at least 30 s. All data mean ± sem.

DOI: https://doi.org/10.7554/eLife.46750.004

The following source data is available for figure 2:

**Source data 1.** Individual behavioural data (movement cycles per time bin) of the foot tapping experiment depicted in panel 2B.
DOI: https://doi.org/10.7554/eLife.46750.005

**Source data 2.** Individual behavioural data (movement cycles per time bin) of the eye movement experiment depicted in panel 2C.
DOI: https://doi.org/10.7554/eLife.46750.006

**Source data 3.** Individual behavioural data (movement cycles per time bin) of the finger tapping experiment depicted in panel 2D.
DOI: https://doi.org/10.7554/eLife.46750.007

---

*length × time* interaction, F(10,272) = 2.329, p=0.012, **Figure 3B**): After long breaks, initial tapping speed was high and, subsequently, strongly diminished when tapping was performed for 30 s. By contrast, after short breaks, the initial tapping speed was clearly reduced but further tapping speed reductions were less pronounced. This demonstrates that break length had a strong influence on the initial tapping speed (0–10 s), but not the final tapping speed (20–30 s). Next, we tested whether shorter breaks allowed for less recovery as reflected by slower tapping speed at the start of the next trial. We calculated a recovery index by subtracting the movement speed before a break from the movement speed immediately after the break (higher index indicates more recovery of tapping speed) for both the slowing and control conditions. We found a significant *condition × break length* interaction (LMEM, F(5,16)=8.771, p<0.001, **Figure 3D**) indicating that longer breaks lead to more recovery than shorter breaks. Interestingly, participants' tapping performance seemed to slightly deteriorate during short breaks after control trials. Note that this effect was not driven by the final tapping speed, as there was no significant difference in tapping speed before the break (**Figure 3C**). Finally, there was a significant correlation between the average amount of slowing observed for an individual participant and the slope of recovery during the break (n = 17, Pearson r = 0.7543, p<0.001, **Figure 3E**).

In summary, our behavioural results show that motor slowing occurs during prolonged tapping at maximal movement speeds irrespective of which effector or tapping task is performed. Less motor slowing is observed when movements are performed at slower speeds or for shorter durations.

**Table 1.** Comparison of movement speed (measured in movement cycles per 10 s) during the first 10 s of the first and last trials for different effectors, as well as movement speed within a 30 s trial (first vs. last 10 s).
The results show that movement speed at the beginning of each trial is stable, whereas movement speed decreased significantly within a trial. All values mean ± sem.

| | Speed during first 10s | | Speed within 30s | | | |
|---|---|---|---|---|---|---|
| | **First Trial** | **Last Trial** | **First 10s** | **Last 10s** | **N Subjects** | **N Trials** |
| 2-finger tapping | 39.48±1.61 | 37.00±1.24 | 37.89±1.12 | 31.82±1.04 | 25 | 16 |
| Foot tapping | 50.00±4.16 | 53.67±4.20 | 55.167±3.25 | 47.233±2.89 | 12 | 20 |
| Eye Movement | 14.33±0.83 | 16.83±0.95 | 15.817±0.77 | 12.933±0.67 | 12 | 20 |

mean ± standard error of the mean, all values Movement Cycles per 10s.
DOI: https://doi.org/10.7554/eLife.46750.012

Similar characteristics have been shown previously for more conventional forms of performance fatigability, for example, when performing isometric contractions (*Dideriksen et al., 2011*). However, our data also suggest that the mechanism which causes slowing for fast, long-lasting repetitive movements appears to quickly recover during the subsequent 25–30 s break following an approximately linear time course.

## Decreased movement speed leads to increased fMRI activation

The same paradigm was performed while fMRI was used to localise which brain areas might be specifically involved in motor slowing (new cohort with n = 25, experiment 6). The fMRI experiment included slowing conditions, control conditions, recovery periods and true rest periods (i.e. periods where participants rested after they had fully recovered, see Materials and methods for further details; *Figure 4A*). In the MR scanner, the participants exhibited significant motor slowing (*Figure 4C*) with a similar effect size to that observed during the behavioural experiment above ($F_{(2,48)}$=85.557, p<0.001, Cohen's d = 1.98). Tapping with the right (dominant) hand activated a typical sensorimotor network (*Figure 4B*, purple; *Supplementary file 2*), including left primary sensorimotor cortex (SM1), bilateral dorsal premotor cortex (PMd), bilateral supplementary motor area (SMA), right cerebellum lobule HVI (Cb), left posterior putamen (Put), left ventrolateral thalamus (Tha), and bilateral secondary somatosensory cortex (S2). To identify areas specifically related to motor slowing, we modelled slowing as a linearly increasing parametric modulator of the tapping condition (*Büchel et al., 1998*). We found that all motor areas showed a trend towards an activation increase even though tapping speed decreased due to motor slowing. However, this effect only reached significance for voxels in contralateral SM1, PMd and SMA (*Figure 4B*, blue; *Figure 4D,E*). We also investigated the 30 s recovery periods following either the 30 s slowing condition or the 10 s control condition. To that end, we modelled recovery as a linear increase during the breaks after the slowing condition, but not after the control condition. We found a significant effect of recovery for voxels in SM1, PMd, SMA ($p_{FWE}$ <0.05; *Figure 4B,D,E* green). Additionally, we found increased activation in the ipsilateral cerebellar motor lobules (HVI) and contralateral S2 associated with recovery during the break. All of these areas showed decreasing activity over the course of recovery which was significantly larger after the slowing condition than after the control condition. Note that performing additional analyses using a block design (i.e. 10 s blocks within each condition) yielded similar results (*Figure 4D,E*).

Thus, somewhat counterintuitively, our fMRI analyses revealed that a reduction in tapping speed during the slowing condition was associated with (i) an activation increase in the motor network which (ii) gradually normalised during the subsequent recovery period.

## Motor slowing leads to electrophysiological after-effects in the alpha-band

It is well-known that the BOLD signal has poor temporal resolution and, thus, we cannot exclude that the effects observed during the recovery period were driven by inaccuracies in modelling the

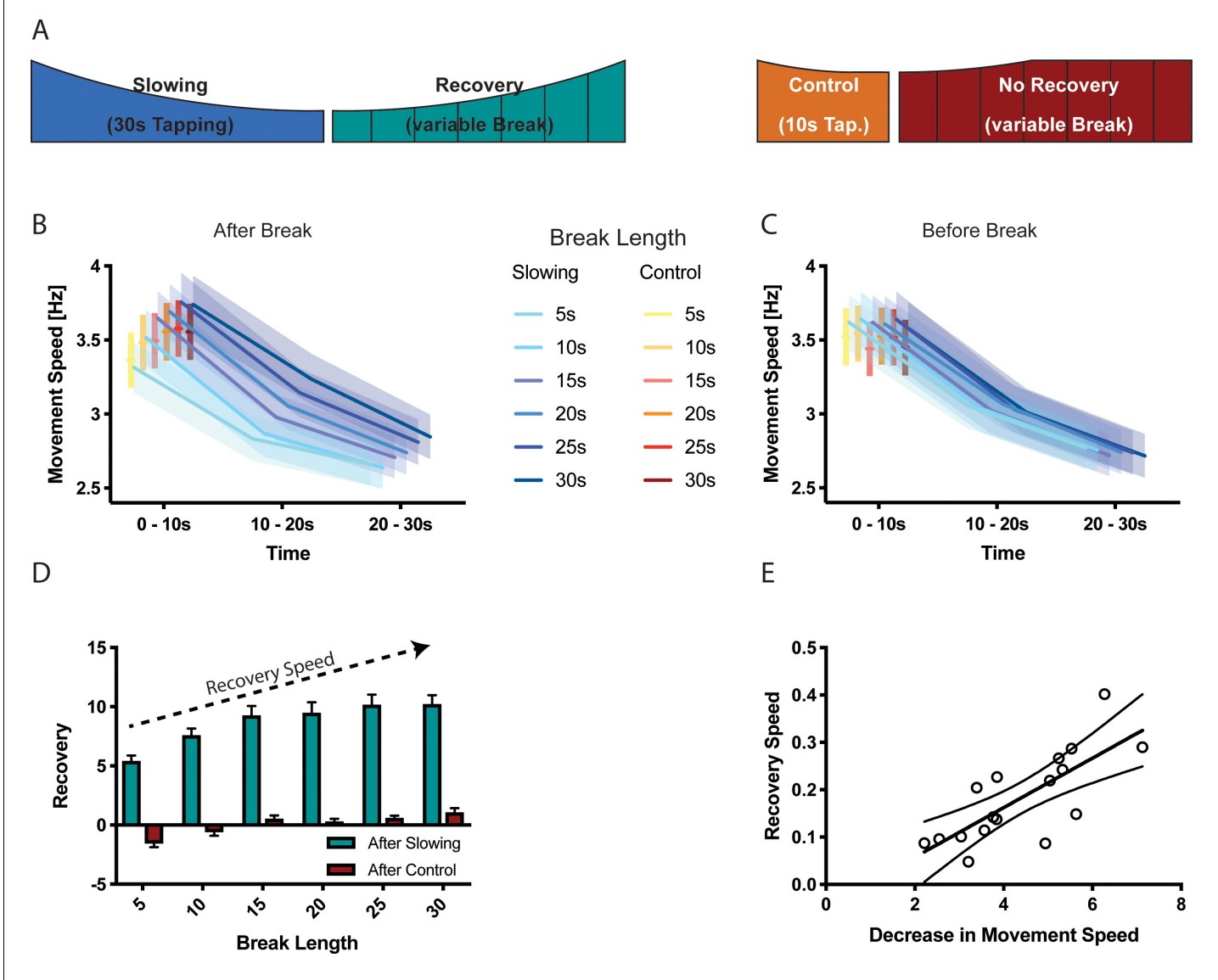

**Figure 3.** Results of the recovery experiment (N = 17). (**A**) Experimental paradigm to characterise the recovery process after motor slowing. Participants were instructed to tap with two fingers for 30 s (slowing condition) or 10 s (control condition) followed by breaks of different duration (5–30 s). (**B**) Initial movement speed was lower when subjects had less time to recover. (**C**) No such pattern was observed before the break. (**D**) Calculating the recovery (i.e. comparing movement speed immediately before and after the break) shows that movement speed recovers within the first 20 s of the break. (**E**) The recovery speed was directly correlated with the decrease in movement speed across participants. All values mean ± sem.

DOI: https://doi.org/10.7554/eLife.46750.008

The following source data is available for figure 3:

**Source data 1.** Individual behavioural data (movement cycles per time bin) after different break lengths depicted in panel 3B.
DOI: https://doi.org/10.7554/eLife.46750.009

**Source data 2.** Individual behavioural data (movement cycles per time bin) before different break lengths depicted in panel 3C.
DOI: https://doi.org/10.7554/eLife.46750.010

**Source data 3.** Individual data of recovery depicted in panel 3D.
DOI: https://doi.org/10.7554/eLife.46750.011

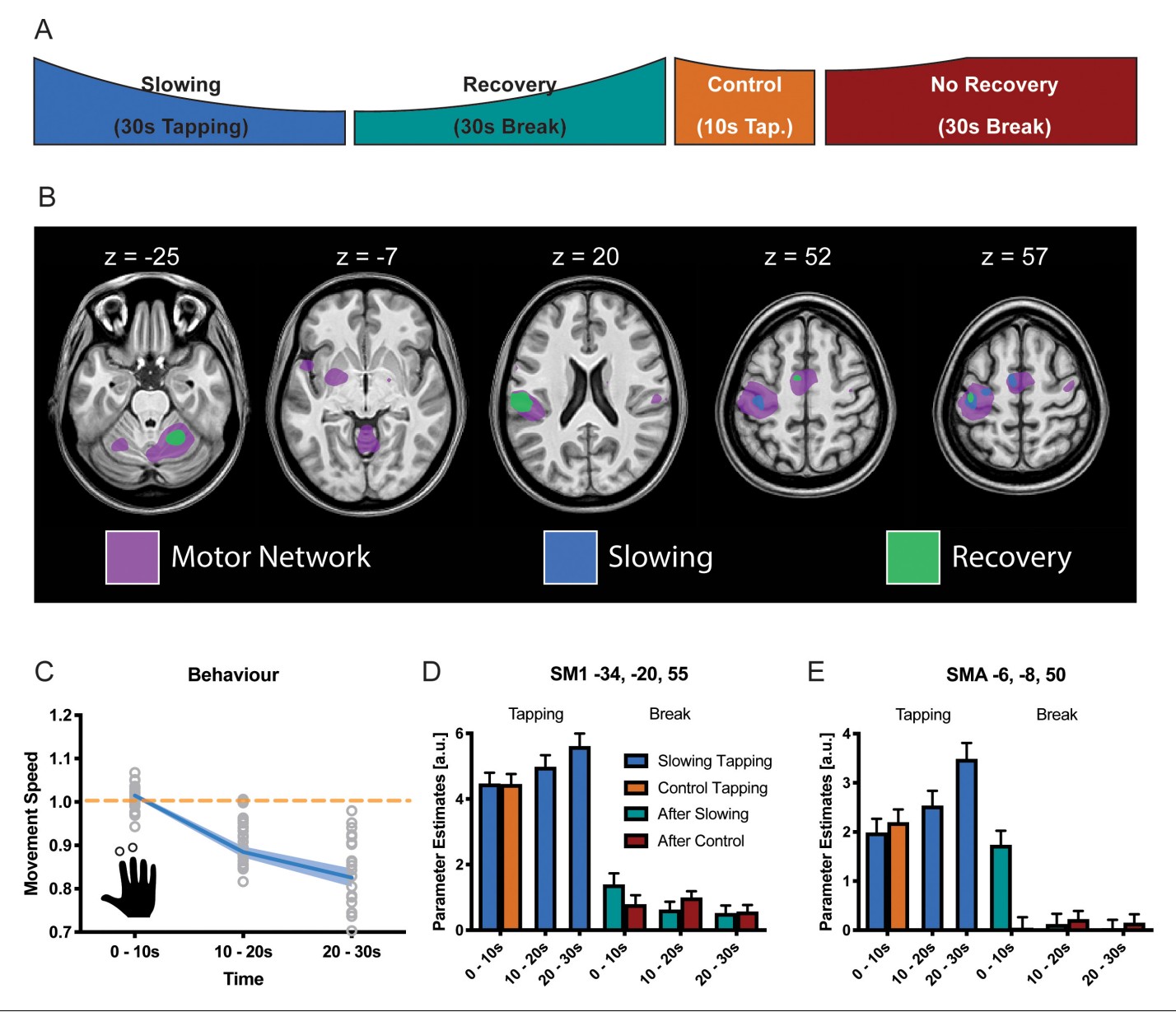

**Figure 4.** Results of the functional magnetic resonance (fMRI) experiment (N = 25). (**A**) Participants performed slowing (30 s) or control (10 s) two-finger tapping followed by a break of 30 s in the fMRI scanner. (**B**) A typical motor network was activated during tapping (pink, $p_{FWE}$ <0.05), however, only areas in primary sensorimotor cortex, premotor cortex and supplementary motor area (SMA) showed increased activity with decreased tapping speed (blue, $p_{FWE}$ <0.05). Additionally, cerebellum and secondary somatosensory cortex show decreasing activation during recovery (green, $p_{FWE}$ <0.05) (**C**) Motor Slowing during the behavioural task (blue line) normalised to control condition (orange line). (**D,E**) Parameter estimates from primary sensorimotor cortex and SMA show increased activity within those areas with decreasing movement speed and subsequent recovery of this effect during the break. All values mean ± sem.

DOI: https://doi.org/10.7554/eLife.46750.013

The following source data is available for figure 4:

**Source data 1.** Individual behavioural data (movement cycles per time bin) of the fMRI experiment depicted in panel 4C.
DOI: https://doi.org/10.7554/eLife.46750.014

**Source data 2.** T-map of fMRI activation (motor network) depicted in panel 4B.
DOI: https://doi.org/10.7554/eLife.46750.015

**Source data 3.** T-map of fMRI activation (slowing) depicted in panel 4B.
DOI: https://doi.org/10.7554/eLife.46750.016

**Source data 4.** T-map of fMRI activation (recovery) depicted in panel 4B.
DOI: https://doi.org/10.7554/eLife.46750.017

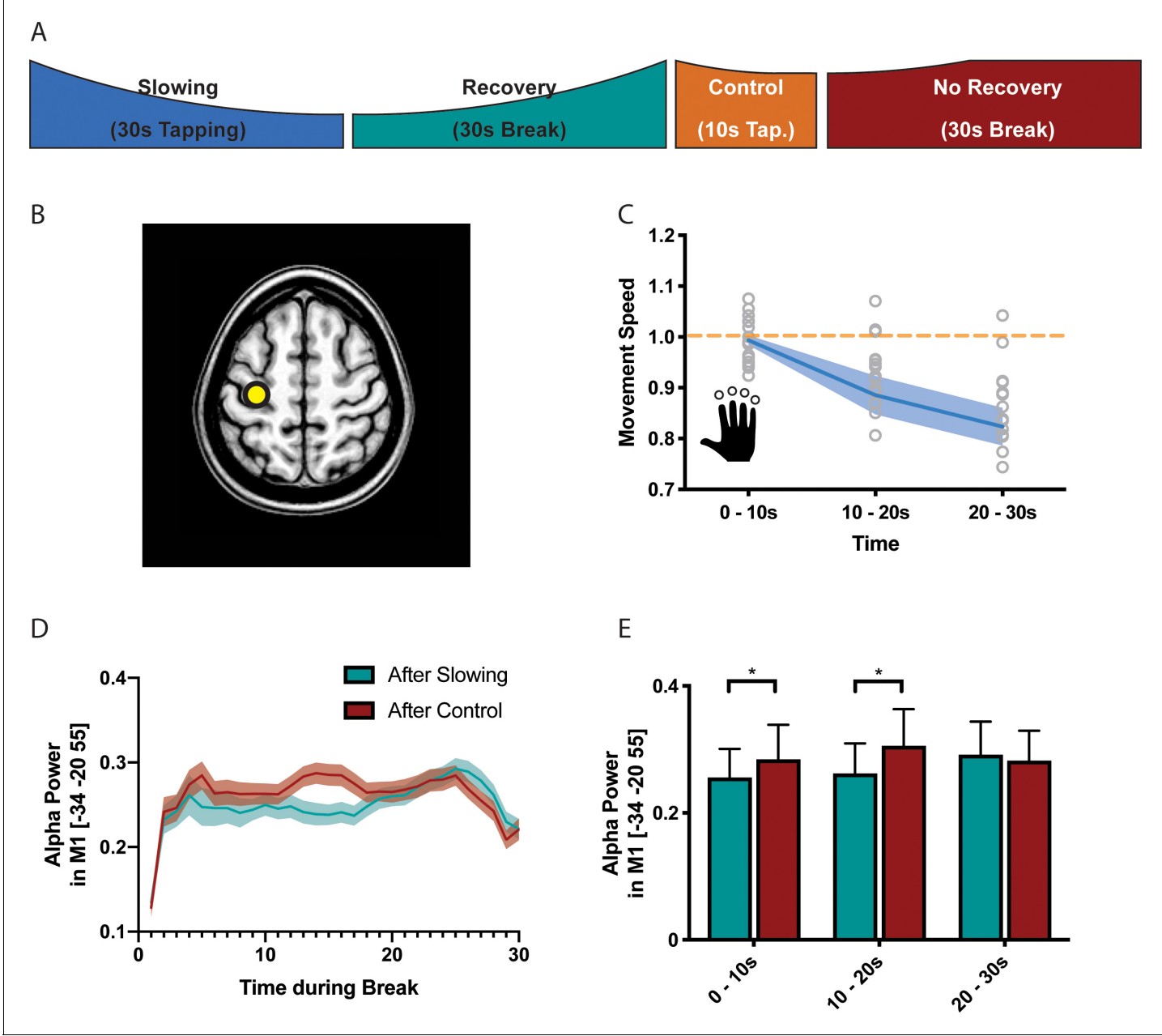

**Figure 5.** Results of the electroencephalography (EEG) experiment (N = 17). (**A**) Participants performed slowing (30 s) or control (10 s) sequence tapping followed by a break of 30 s while EEG was measured. (**B**) Source localisation was performed using eLoreta and fluctuations of alpha power extracted from primary sensorimotor cortex (SM1). (**C**) Motor slowing during the behavioural task (blue line) normalised to control condition (orange line). (**D**) Power over time during breaks after tapping (green = break after slowing tapping, red = break after control tapping, dotted lines = sem.). (**E**) Same as (**D**), binned in 10 s blocks for statistical analysis. All values mean ± sem. *p<0.05.

DOI: https://doi.org/10.7554/eLife.46750.018

The following source data is available for figure 5:

**Source data 1.** Individual behavioural data (movement cycles per time bin) of the EEG experiment depicted in panel 5C.
DOI: https://doi.org/10.7554/eLife.46750.019
**Source data 2.** Individual alpha power data depicted in panel 5E.
DOI: https://doi.org/10.7554/eLife.46750.020

individual hemodynamic response function (*Balsters and Ramnani, 2011*; *Handwerker et al., 2012*). Therefore, we performed a separate experiment (n = 17, experiment 7) where we measured high-density EEG during the recovery period following either the slowing condition (30 s tapping) or the control condition (10 s tapping; *Figure 5A*). Again, we found a significant behavioural effect of motor slowing (*Figure 5C*, F(2 and 36)=14.796, p<0.001, Cohen's d = 1.04). The EEG analysis focused on neuronal oscillations in the alpha (8–14 Hz), beta (14–30 Hz) and gamma (30–40 Hz) band, that is cortical rhythms which have been associated with motor control (*Cheyne, 2013*; *Pfurtscheller, 1992*; *Pogosyan et al., 2009*; *Ritter et al., 2009*). We first performed source localisation using eLORETA (*Pascual-Marqui et al., 2011*) and extracted the power envelopes from three seed regions in SMA (MNI −6 −8 50), left PMd (MNI −28 16 70) and left SM1 (MNI −34 20 55), that is those areas that were identified by the fMRI experiment and exhibited a significant activation increase for decreasing tapping speed (*Figure 5B*). For SM1 (but not PMd and SMA), we found that event-related power synchronisation in the alpha band (*Pfurtscheller et al., 1996*) was more strongly decreased immediately after the slowing condition than after the control condition (*Figure 5D*, green vs. red). To further quantify this differential recovery process, we averaged alpha-power within three time bins of 10 s each after the break (*Figure 5E*) and performed a linear mixed effects analysis with the factors condition (slowing vs. control), time (during break, that is the three bins), and trial (to check for changes in alpha over the whole experiment). We found a significant *condition x time* interaction (F(2,1136) = 3.195,p=0.041) for left SM1. Post-hoc comparisons revealed that alpha-power was significantly lower during the first two time bins (0–10 s and 10–20 s) of the recovery period after the slowing condition than after the control condition ($p_{uncorr}$ <0.05), confirming that alpha power recovered more quickly after the control condition than after the slowing condition. No such differences in the time course of recovery were observed for the beta or gamma band. This finding is interesting because it provides the first experimental evidence that recovery from motor slowing in SM1 can be detected with neurophysiological measurements applied when the participant is at rest. Note that, unlike fMRI, EEG offers a high temporal resolution which allowed us to accurately dissociate the tapping conditions from the subsequent break periods where no overt motor activity was observed. Our EEG results are consistent with the fMRI findings (see above) since it has been shown that low alpha power within the sensorimotor system is associated with an elevated BOLD signal (*Bächinger et al., 2017*; *Hipp et al., 2012*; *Ritter et al., 2009*). In line with these findings, it has been proposed that activity in the alpha band reflects top-down inhibitory control processes (*Klimesch et al., 2007*) suggesting that low alpha power - as observed in SM1 immediately after the slowing condition - reflects a prominent release of inhibition which gradually recovered over the time course of the break. While alpha power was also suppressed immediately after the control tapping condition, it recovered much quicker. Thus, our EEG experiments corroborate the fMRI results by suggesting that (i) after-effects of motor slowing can be measured during the first 10 s of the recovery period and (ii) recovery from motor slowing is associated with re-establishing inhibitory activity in SM1.

## Motor slowing is associated with a release of inhibition in SM1

Whilst the alpha-band has been associated with inhibitory control, EEG can only reveal indirect insights into the activity of inhibitory circuits in SM1. We therefore performed a follow-up experiment (n = 13, experiment 8) and directly probed the activity of $GABA_A$ circuits by applying a TMS short-interval intracortical inhibition (SICI) protocol during the breaks following either the slowing condition (30 s tapping of an over-learned 4-element sequence) or the control condition (10 s of the same tapping task; *Figure 6A*) (*Kujirai et al., 1993*; *Ziemann et al., 1996*). Again, there was a significant decrease of the movement speed during the slowing condition (F(3,36)=42.94, p<0.001, Cohen's d = 2.40; *Figure 6B*). In this experiment, we measured the effect of slowing versus the control condition on two separate days to limit the overall duration of each experimental session. We performed several control analyses to ensure that both the behavioural and the electrophysiological measurements were comparable between the sessions. First, tapping speed for the first 10 s bin was similar and not significantly different between sessions (paired t-test: t(12)=1.303, p=0.217). Second, for both sessions, rest motor threshold (RMT), conditioning stimulus intensity (CS) and test stimulus intensity (TS) were similar and not significantly different (see *Supplementary file 3*). Finally, SICI measured at rest prior to the tapping conditions (Pre measurements) was similar and not significantly different between the sessions (*Figure 6C*, *Slowing vs. Control at Pre*: F(1,12)=0.086, p=0.775).

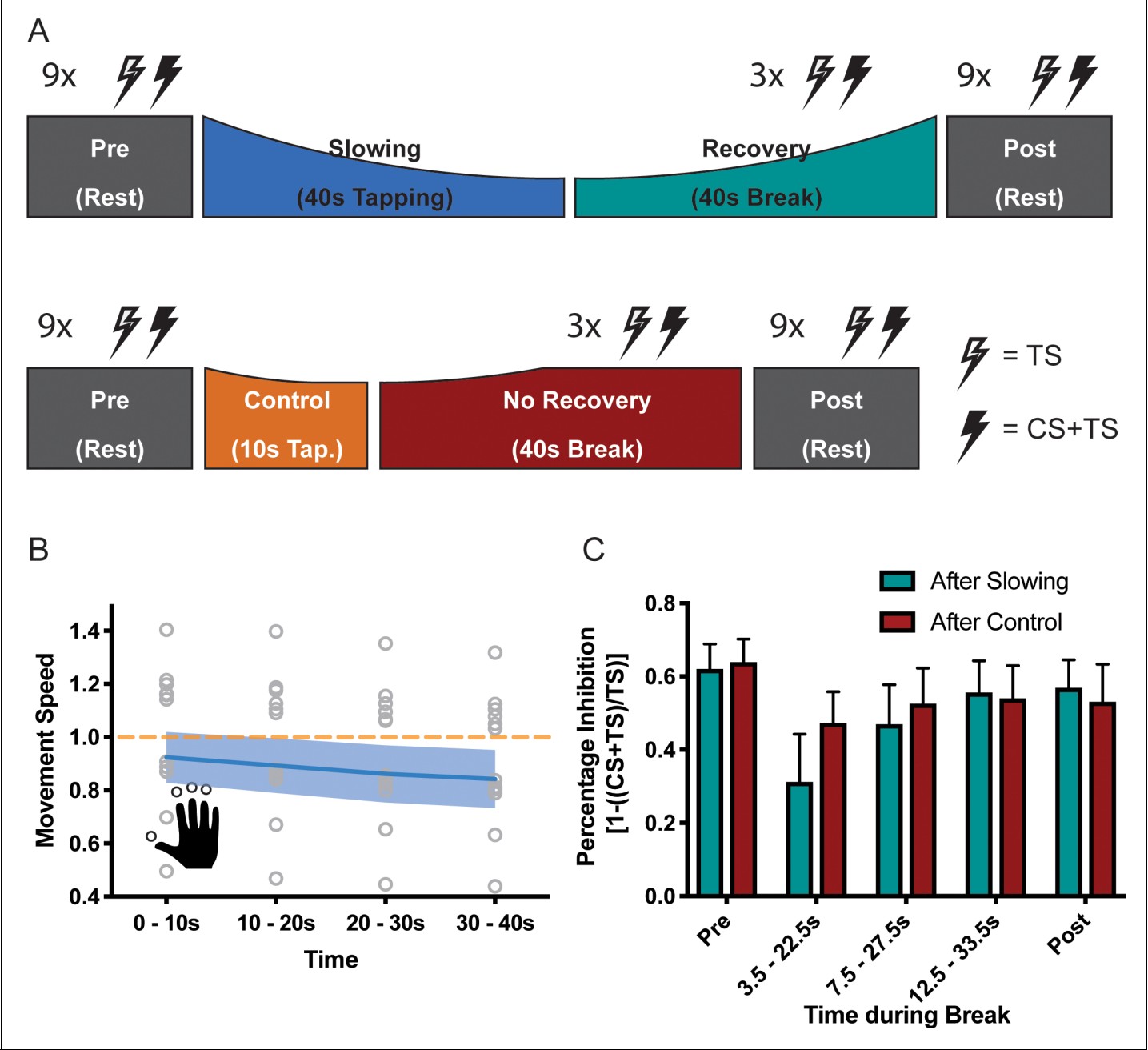

**Figure 6.** Results of the short latency intracortical inhibition (SICI) experiment (N = 13). (**A**) SICI was measured before (Pre) and after (Post), as well as during the break after either slowing (40 s) or control (10 s) tapping. (**B**) Motor slowing during the behavioural task (blue line) normalised to control condition (orange line). Note that in this experiment the two conditions were tested on two different days, therefore, there was a difference in initial speed across sessions. (**C**) Percentage Inhibition of primary motor cortex before (Pre), for the first (0–10 s), second (10–20 s) and third (20–30 s) conditioning stimulus during the break, and at the end of the session (Post) where participants executed either the slowing (green) or control (red) tapping condition. It can be seen that slowing leads to disinhibition of primary motor cortex immediately after tapping. All values mean ± sem.
DOI: https://doi.org/10.7554/eLife.46750.021

The following source data is available for figure 6:

**Source data 1.** Individual behavioural data (movement cycles per time bin) of the SICI experiment depicted in panel 6B.
DOI: https://doi.org/10.7554/eLife.46750.022
**Source data 2.** Individual SICI data depicted in panel 6C.
DOI: https://doi.org/10.7554/eLife.46750.023

Comparison of SICI before (Pre) and after (Post) the behavioural paradigm revealed only a minimal decrease in inhibition over the course of the experiment, which was highly similar between sessions (*Figure 6B*, *Time (Pre vs Post)*: F(1,12)=1.950, p=0.188, *Time (Pre, Post) x Condition* interaction: F(1,12)=0.214, p=0.652). Importantly, we found that recovery of SICI during the break followed different time courses when measured after the slowing versus the control condition (*Figure 6C*), which was statistically confirmed by a significant *condition x time* interaction (F(4,12)=5.573, p=0.009). More specifically, SICI was strongly decreased immediately after the motor slowing condition (0–10 s of recovery period) as compared to both the Pre and Post measurements. However, it returned back to baseline at the end of the recovery period (20–30 s) (*Figure 6C*, green bars). By contrast, after the control condition, SICI was only slightly decreased and recovered almost immediately after tapping (*Figure 6C*, red bars). Importantly none of these effects were driven by changes in background EMG (see *Supplementary file 4*).

Thus, in line with the fMRI and EEG findings reported above, the TMS experiment revealed further evidence that performing repetitive movements for a period of $\geq$ 30 s leads to a strong release of inhibition within SM1 that gradually normalised over time.

## Motor slowing is associated with decreased surround inhibition and increased coactivation of antagonistic muscles

How can this release of intracortical inhibition be reconciled with the observation that repetitive movements become slower? Repetitive movements in general rely on precise timing between agonistic and antagonistic muscle activity: whenever the agonistic movement is performed, corresponding antagonistic motor activity needs to be suppressed and vice-versa. Accordingly, the observed increase in excitability in the motor system might be 'maladaptive' and we hypothesised that it might indicate a breakdown of surround inhibition. Surround inhibition in the motor system describes the phenomenon that selective preparation of, for example, an index finger movement, decreases excitability of surrounding fingers (*Beck and Hallett, 2011*). Applied to motor slowing, one would expect that surround inhibition of antagonistic movements should be more strongly diminished after slowing than after the control condition, and this effect should be observable in form of (i) a gradual increase of coactivation during the slowing condition; and (ii) reduced surround inhibition when measured immediately after the slowing condition (i.e. during the first 10 s of the recovery period) with TMS. To test these predictions, we performed a final experiment (n = 19, experiment 9) where participants performed repetitive thumb movements for either 30 s (slowing condition) or 10 s (control condition; *Figure 7A*). Again, we found significant motor slowing (F(2,36 = 21.484, p<0.001, Cohen's d = 1.15)). Electromyography (EMG) was measured from the thumb flexor abductor pollicis brevis (APB) and its antagonist, that is the extensor pollicis longus (EPL) during movement and rest. Muscle coactivations were assessed by calculating the overlap between the rectified APB and EPL EMG signals (*Figure 7B*). We found a significant increase in coactivation over the course of tapping (*Figure 7C*, F(2 and 36)=9.915, p=0.001), and over the course of motor slowing, changes in coactivation in a single participant was directly related to his/her changes in movement speed (LMEM, F(1,1561.414) = 4.243, p=0.040). We did not find such an association for any other EMG parameter (i.e. amplitude, frequency of individual muscles). Additionally, we quantified surround inhibition during the recovery phase immediately after the slowing versus control condition. These measurements took place during two separate sessions where participants were instructed to perform a thumb abduction which elicited an EMG-triggered TMS pulse. TMS was either triggered immediately (i.e. 3 ms after movement onset $TMS_{Mov}$) or 2 s after movement onset ($TMS_{Con}$). The quotient of the motor evoked potentials elicited by the two pulses served as a measure for surround inhibition (see Materials and methods for details). Again, we made sure that the surround inhibition measurements were comparable across sessions. First, tapping speed during the first 10 s bin was comparable between sessions and not significantly different (paired t-test t(18) = 1.381, p=0.184). Second, rest motor threshold was similar across sessions and the size of $TMS_{Con}$ did not change between the two sessions or within the different timepoints of the break (see *Supplementary file 5*). To compare the results between the two sessions we normalised surround inhibition measurements obtained during the break to the individual Pre measurements. Normalised surround inhibition measured during the first 10 s of the recovery phase was significantly decreased for the slowing condition compared to the control condition, but reached similar levels at the end of the break (significant *condition x time* interaction (*Figure 7E*; F(2,18) =

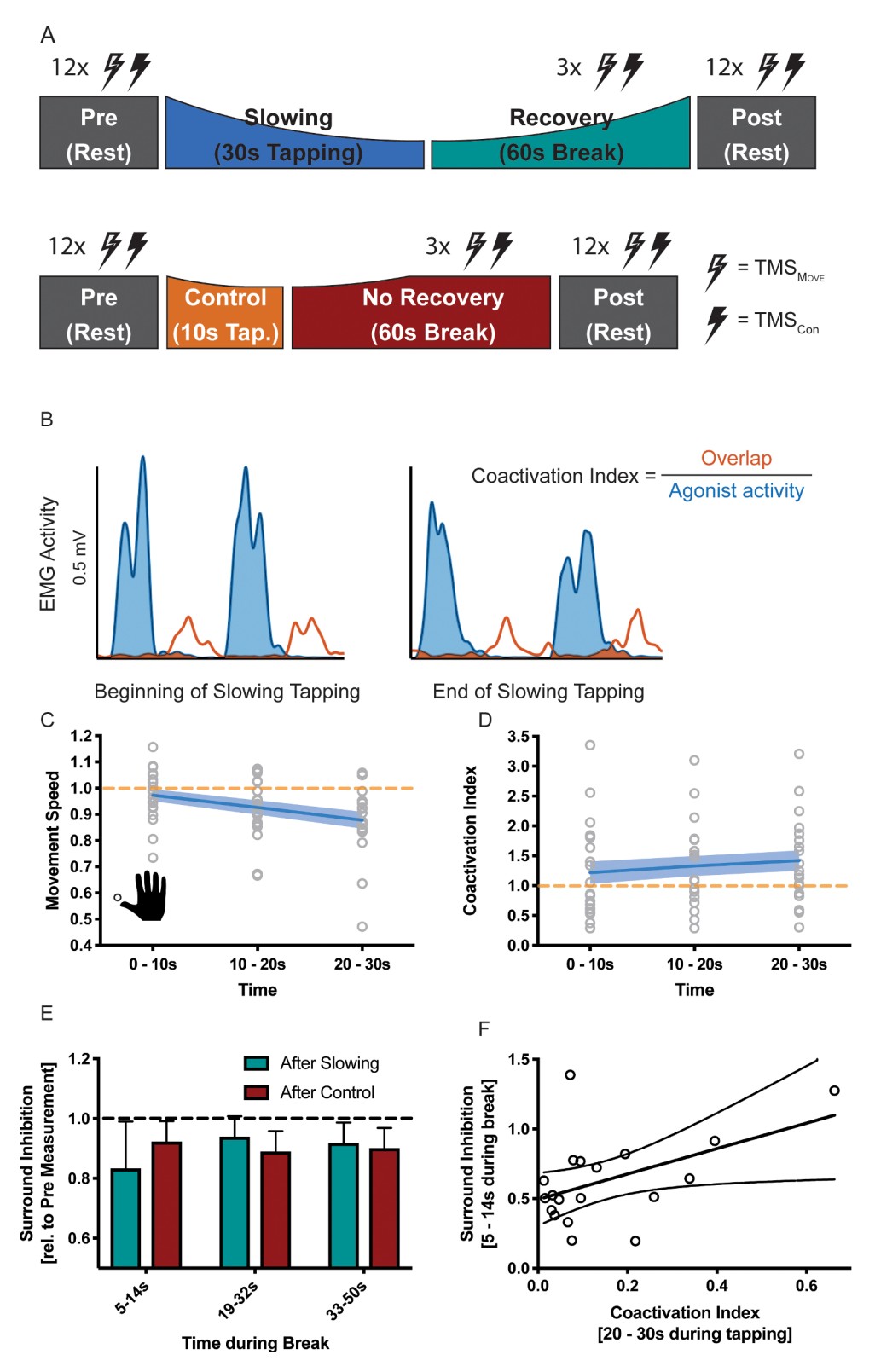

**Figure 7.** Results of the surround inhibition experiment (N = 19). (**A**) Surround inhibition was measured before (Pre) and after (Post), as well as during the break after either slowing (30 s) or control (10 s) tapping. The length of the break was increased to 60 s to have enough time for the measurements. (**B**) The amount of coactivation between agonistic and antagonistic muscles was calculated by dividing the overlap between the smoothed and rectified EMG of the two muscles normalised to the area under the curve of the agonist. (**C**) Movement speed decrease in the surround inhibition experiment.
*Figure 7 continued on next page*

*Figure 7 continued*

Note that in this experiment the two conditions were tested on two different days, therefore, there was a difference in initial speed across sessions. (D) Increase in coactivation index between APB and EPL over the course of motor slowing. (E) Difference in surround inhibition normalised to Pre-measurements. Surround inhibition is decreased immediately after motor slowing and returns to baseline over the course of the break. (F) The amount of coactivation immediately before the break predicted the amount of surround inhibition at the beginning of the break (R2 = 0.23, p=0.039). Note that surround inhibition in panel (F) is the absolute amount of surround inhibition, whereas it is normalised to the Pre measurement in panel (E). All values mean ± sem.

DOI: https://doi.org/10.7554/eLife.46750.024

The following source data and figure supplement are available for figure 7:

**Source data 1.** Individual behavioural data (movement cycles per time bin) of the surround inhibition experiment depicted in panel 7C.

DOI: https://doi.org/10.7554/eLife.46750.026

**Source data 2.** Individual coactivation data of the surround inhibition experiment depicted in panel 7D.

DOI: https://doi.org/10.7554/eLife.46750.027

**Source data 3.** Individual surround inhibition data depicted in panel 7E.

DOI: https://doi.org/10.7554/eLife.46750.028

**Figure supplement 1.** Raw MEP Ratios from the surround inhibition experiment for FDI (A) and ADM (B).

DOI: https://doi.org/10.7554/eLife.46750.025

3.908, p=0.039)). Note that this effect was only present for the FDI muscle, but not the ADM muscle (see *Figure 7—figure supplement 1*). Further, across participants, the average amount of coactivation in the last 10 s of slowing trials was predictive of the average amount of surround inhibition measured early during the recovery period such that individuals with a high coa index exhibit a strong release of surround inhibition (*Figure 7F*; linear regression model; $R^2$ = 0.228, p=0.039). Again, we confirmed that the observed effect was not driven by background EMG (see *Supplementary file 6*). Taken together, these results suggest that the amount of motor slowing, the amount of coactivation between the agonistic and antagonistic muscle, and the strong release of surround inhibition are associated. We therefore propose that fast repetitive movements cause an increase of the excitation-inhibition ratio at the level of M1 which results, at least partly, from a breakdown of surround inhibition. Accordingly, coactivation of agonistic and antagonistic muscles increases, which ultimately leads to a decrease in movement speed as observed during motor slowing.

## Discussion

We demonstrate that fast repetitive movements are subject to a gradual reduction of movement speed or 'motor slowing', even when the required forces are clearly submaximal and each single contraction is brief. Importantly, we show that motor slowing is a continuous process which starts immediately with movement onset and critically depends on the initial movement speed. The motor slowing phenomenon was replicated across nine different cohorts (consisting of 157 participants in total) and statistics consistently revealed large effect sizes (all Cohen's d > 1.04) irrespective of whether movements were performed with fingers, feet or eyes, or whether the motor task was simple (e.g. single joint movements) or more complex (e.g. overtrained four-element sequence). Moreover, motor slowing recovered quickly during the subsequent break with a linear relationship between the rate of slowing and the rate of recovery (*Figure 3*). The latter finding is particularly important since it suggests that circuits which mediate slowing might exhibit measurable after-effects during the subsequent break. This offers a unique opportunity to disambiguate the mechanisms of motor slowing from neural activity related to movement execution per se. In summary, in line with previous work on fatigability of repetitive movements (*Arias et al., 2015*; *Madrid et al., 2016*; *Rodrigues et al., 2009*; *Teo et al., 2014*; *Teo et al., 2012b*), our findings suggest that motor slowing is a robust phenomenon which reflects a general organisational principle of the motor system (*Viviani and Cenzato, 1985*). Note that the motor slowing phenomenon depends on both the initial tapping speed and the duration that tapping must be maintained. Our principle paradigm varied the *time period* of fast tapping, that is >30 s, which evoked substantial motor slowing versus 10 s, which evoked only minor slowing and served as a control condition. However, experiment 2 (*Figure 1B*) indicates that analogous effects would have been obtained if the overall tapping

duration had been kept constant while initial tapping speed was either high (evoking substantial motor slowing) or low (evoking only minimal slowing).

Our design was motivated by classical experiments investigating how performance fatigability in the motor system manifests for isometric voluntary contractions as a function of time (*Taylor and Gandevia, 2008*; *Gruet et al., 2013*; *Taylor et al., 2016*). For isometric contractions, performance fatigability is defined as a gradual reduction in output force that strongly depends on the initial force level and the duration it must be maintained (*Taylor et al., 2016*). Importantly, the systematic reduction of force output over time is strongest if initial target force levels are high but depends predominantly on peripheral factors (*Place et al., 2010*; *Enoka and Stuart, 1992*). Here we demonstrate an analogous phenomenon for sustained fast repetitive movements which cause performance fatigability even though the force level of a single movement is too low and overall execution times are too short to evoke spinal or neuromuscular mechanisms as shown by previous research (*Rodrigues et al., 2009*; *Miller et al., 1993*). More specifically, it has been suggested that supraspinal mechanisms are the main contributors to motor slowing as investigated here, since it causes neither a change in isometric maximal voluntary contraction (MVC) force, nor a change in force production evoked by electrical stimulation of the muscle (*Arias et al., 2015*; *Madrid et al., 2018*; *Miller et al., 1993*; *Rodrigues et al., 2009*). As such, motor slowing is an interesting paradigm for investigating which central mechanisms might contribute to the more general phenomenon of performance fatigability.

## Motor slowing is associated with increased neural activity within the motor network and, particularly, motor cortex

Our fMRI results revealed a general increase in neural activity within the general cortico-subcortical motor network, with the largest effects observed in PMd, SMA and SM1. No evidence suggested a reduction in neural activity in any motor or non-motor area. The finding of *increased* neural activity in the whole motor network despite *decreased* tapping speed is surprising since previous research has shown an increase in BOLD signal in most motor areas in response to increased sensorimotor activity. More specifically, the change in BOLD signal in S1/M1, premotor cortex, SMA, thalamus and selected basal ganglia nuclei is higher for high levels of force production compared to low levels of force production (*Spraker et al., 2007*). Similarly, performing passive movements with increasing velocity causes a linear increase in the BOLD signal in S1/M1, secondary somatosensory cortex (S2), and SMA, which corresponds well to the rate encoding principles shown for sensory afferent signals (*Dueñas et al., 2018*).

However, our finding is in line with previous work demonstrating increasing activity in SM1, PMd and SMA during fatiguing maximal isometric contractions, as characterised by a progressive decline in maximal finger abduction force (*Post et al., 2009*). Although fatigability of isometric contractions arises mainly at the level of the muscle, there is also a central contribution. This has been shown by the 'superimposed twitch' method, that is applying electrical stimulation which evokes an increment in force (*Gandevia et al., 1996*; *Schillings et al., 2003*). During fatiguing contractions, the amplitude of the superimposed twitch increases gradually suggesting that, despite the observed increase in the BOLD signal of cortical motor areas, the central drive is insufficient to maintain truly maximal contractions (*Post et al., 2009*). We observed the same pattern of results for a repetitive task where muscular or spinal fatigue mechanism play a minor role (*Rodrigues et al., 2009*), confirming that – even though paradoxical at first sight - an increase of net activity within cortical motor areas might underpin a central mechanism mediating performance fatigability.

Our fMRI study was designed to disambiguate changes in BOLD response related to recovery from motor slowing (i.e. immediately after tapping) and true rest periods that were acquired after recovery was complete (i.e. >30 s after tapping, as suggested by *Figure 3*). Interestingly, for various motor areas we observed that the BOLD response remained elevated during the first 10 s after the motor slowing condition. By contrast, the BOLD signal returned much more rapidly to baseline after the control condition. Interestingly, only areas which tended to exhibit an increase in BOLD response during long-lasting tapping gradually reduced their activity during the subsequent break, while no other brain area exhibited a significant change in activation during the break. Note that the overall activation changes of the motor-network under FWE-correction was smaller during tapping than during the break. This higher sensitivity of the fMRI data during the break might result from less inter-subject variability at rest. In addition to our behavioural finding that shortening the break, that is

disrupting the recovery process after slowing with another block of tapping, has clear behavioural consequences for subsequent tapping trials, our fMRI data suggest that the after-effects of slowing are mediated by the same neural substrate as motor slowing itself.

One general concern regarding fMRI is its low temporal resolution, and the after-effects observed during the first 10 s following the motor slowing condition might simply reflect a methodological artefact due to an inaccurate model of the hemodynamic response function. Therefore, we investigated the after-effects of motor slowing versus (less-slowing) control tapping with EEG, which offers limited spatial but excellent temporal resolution. It has been previously been shown that alpha power and BOLD activation in sensorimotor cortices are anticorrelated (*Ritter et al., 2009*). Our EEG data confirmed differential temporal dynamics of alpha-band activity in SM1 during the break, which took longer to recover immediately after the motor slowing condition than after the control condition (*Figure 5*). This suggests that the differential recovery of the BOLD signal is not solely a hemodynamic artifact. Moreover, event-related alpha synchronisation does not differ between long versus short isometric contractions (*Cassim et al., 2000*), therefore, the differential dynamics of alpha-band activity in our EEG study are unlikely the result of different movement durations for the motor slowing versus the (less-slowing) control condition.

In summary, our results are in line with the concept that motor slowing is a form of central fatigability which can be robustly measured in cortical motor areas and, particularly M1, where it outlasts the movement execution phase, a phenomenon that has been previously demonstrated for isometric force production tasks (*Taylor and Gandevia, 2008*).

## Motor slowing is associated with the release of surround inhibition of primary motor cortex

Central fatigability has been conceptualised as a decrease in voluntary drive, resulting in descending motor commands that are insufficient to maintain high tapping speed or high isometric muscle contractions (*Gandevia et al., 1996*; *Kluger et al., 2013*). Yet, we and others (*Post et al., 2009*) revealed that central fatigability is associated with an increase of neural activity as measured with fMRI. How can these paradoxical findings be reconciled? Both EEG (*Figure 5*) and SICI results (*Figure 6*) suggest a decrease of inhibition within M1 shortly after motor slowing. More specifically, EEG measurements revealed that alpha was strongly reduced immediately after tapping, and recovered significantly slower in the first 20 s after motor slowing compared to the control condition. High alpha-activity has been suggested to reflect inhibitory activity (*Klimesch et al., 2007*), thereby providing indirect evidence that motor slowing is associated with a pronounced release of SM1 inhibition, which is gradually restored during the subsequent break. These EEG results were further supported by SICI measurements that assess the activity of GABA$_A$-ergic networks within M1 (*Werhahn et al., 1999*). We found a reduction in SICI indicating the release of inhibition in M1, which was most pronounced during the first 10 s after the slowing condition (i.e. for the first SICI pulse) but was gradually restored during the 30 s break (*Figure 5*). Under the assumption that these after-effects of reduced inhibition are representative of neurophysiological changes underpinning the motor slowing phenomenon during movement execution, the increase in BOLD signal is likely to reflect higher net excitation of the motor network (*Waldvogel et al., 2000*), resulting from a shift of the excitation-inhibition balance towards excitation (*Logothetis, 2008*). But how could an increase of the excitation-inhibition ratio within M1 cause decrements in tapping performance? Repetitive single joint movements require the sequential activity of agonistic and antagonistic muscles. This alternating activation pattern needs to be particularly well-timed for fast, repetitive movements requiring that the agonist is excited while the antagonist is inhibited to minimise muscular coactivation. Furthermore, it has been shown that the ability to selectively activate one specific muscle while suppressing unwanted activity in other muscles requires surround inhibition (*Beck and Hallett, 2011*). Here we demonstrate that motor slowing is associated with a gradual increase of coactivation between antagonistic muscles, and that this change of coactivation predicts the amount of motor slowing observed across participants. We further found that surround inhibition was decreased immediately after tapping, which was significantly stronger following the motor slowing condition than the control condition and gradually recovered during the break. Importantly, individuals which exhibited strong coactivation during the last 10 s of the motor slowing condition also exhibited low surround inhibition during the first 10 s of the subsequent break, as indicated by a significant association between these phenomena. These observations strongly suggest that motor slowing is at least

partially related to the breakdown of surround inhibition which is, in turn, associated with an increase of coactivation between antagonistic muscle groups, thereby making repetitive movements increasingly effortful and slow. Even though we tested healthy participants, it is possible that motor slowing shares mechanistic similarities with pathological forms of fatigue that are frequently observed in neurological patients (*Lewis and Wessely, 1992*; *Lou, 2009*; *Ranjith, 2005*; *Watanabe et al., 2008*). Our study provides the testable hypothesis that an abnormal release of inhibition within SM1 might be related to pathological fatigability and/or bradykinetic movements.

## Potential neuronal basis of motor slowing at the microscopic level

One open question is how our findings - which were all obtained at the macroscopic level - relate to activity at the cell level. It is well known that neurons in primary motor cortex are tuned to represent movement direction in extrinsic space (*Georgopoulos and Carpenter, 2015*; *Georgopoulos et al., 1982*; *Georgopoulos et al., 1986*) and that this tuning is sculpted by inhibitory mechanisms (*Merchant et al., 2008*). In particular, it has been proposed that circuits mediating local inhibition lead to a sharpening of the directional tuning curve, which determines the accuracy of the directional motor command (*Mahan and Georgopoulos, 2013*). Although this theory was first discussed with respect to the speed-accuracy trade-off of single movements, a similar mechanism might play an important role during motor slowing. Here we propose a model of simple flexor-extensor movements similar to experiment 9 (*Figure 7*), which contains two populations of pyramidal cells ($P_{Flex}$ and $P_{Ext}$; *Figure 8*) that are tuned in opposite directions. Inhibitory interneurons shape the width of each tuning curve (I; *Figure 8*). Additionally, the two populations mutually inhibit each other reflecting the mechanism of surround inhibition. At the beginning of tapping, inhibition is strong and the two tuning curves are 'sharp'. However, when the fast tapping needs to be maintained over a longer period of time, surround inhibition breaks down and the tuning curves of both populations become broader (*Figure 8*; right side). In turn, the descending motor command is less accurate making the muscle activation pattern less efficient activating antagonistic muscle groups in parallel. Note that although *Figure 8* shows a direct interaction between the two populations, the broadening of tuning curves might also occur due to a general release of inhibition, which might be controlled by an upstream area, for example, via afferents from premotor cortex or SMA. This model might also explain why brain activity increases during fatiguing isometric contractions (*van Duinen et al., 2007*). Potentially, in the isometric case, the development of a specific force level requires a precisely tuned population of neurons to maintain synergistic control of the muscles involved in the movement.

In summary, our model suggests that a release of inhibition and, particularly, a breakdown of surround inhibition in M1 might be associated with performance fatigability rather than a compensatory mechanism to overcome the reduction in muscular output. In line with this proposal, it has been shown that isometric contractions can be maintained longer when an external focus of attention is adopted (*Kuhn et al., 2017a*). Interestingly, this improvement of performance was accompanied by an increase in SICI (*Kuhn et al., 2017a*; *Kuhn et al., 2017b*) and an increase in surround inhibition (*Kuhn et al., 2018*).

## Conclusion

Here we show that fast repetitive movements are subject to gradual slowing even though each single muscle contraction is brief and submaximal. Based on converging evidence from behavioural, fMRI, EEG and TMS measurements we argue that motor slowing arises from transient neurophysiological changes of supraspinal areas indicating that this form of motor fatigability is largely mediated by central mechanisms. Specifically, we show that motor slowing is accompanied by a gradual shift of the excitation-inhibition balance within primary motor cortex towards more net excitation. Even though paradoxical at first, we show that this shift results from the release of inhibition in M1 and, particularly, the breakdown of surround inhibition causing increased cocontraction between antagonistic muscle groups which, in turn, leads to more and more effortful and slow tapping movements. We further propose a model in which this breakdown of surround inhibition causes a broader tuning of neuronal populations in M1 that encode movement direction, resulting in a sub-optimal and less efficient descending motor command than at the beginning of tapping. Given that motor slowing is not only generalisable across muscle groups and tasks, but is also present in almost all participants,

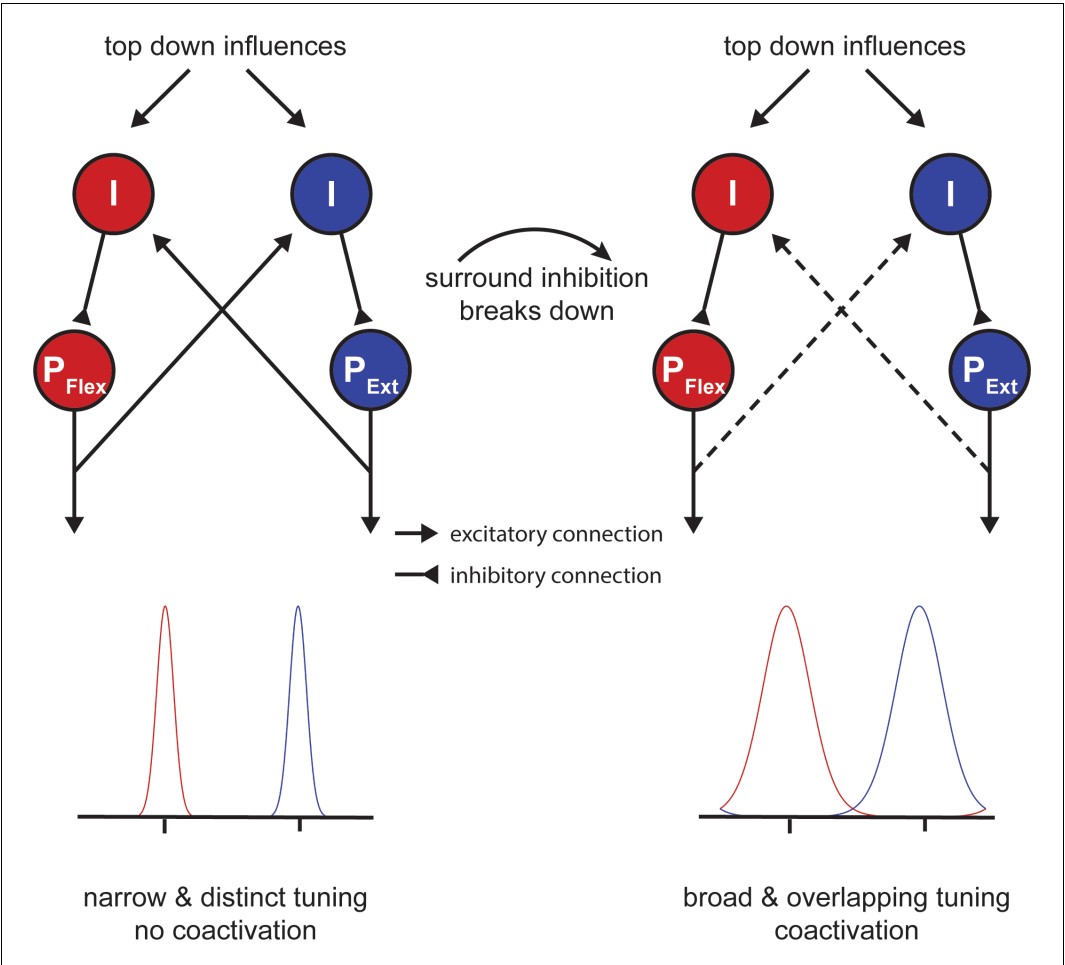

**Figure 8.** Potential mechanism of motor slowing. Two populations of pyramidal neurons (P) control agonistic ($P_{Flex}$) and antagonistic ($P_{Ext}$) movements. The tuning curve of those neurons is under control of inhibitory interneurons (I). At the beginning of tapping inhibition is strong leading to sharp tuning curves and consequently distinct movement activation patterns. Over the course of tapping inhibition breaks down and consequently the tuning curves become broader leading to overlapping activation patterns (i.e. coactivation). Note that the breakdown of inhibition here is shown as a direct input, however, inhibition could also break down due to a reduction of afferent excitatory input to the inhibitory interneurons.
DOI: https://doi.org/10.7554/eLife.46750.029

it reflects a fundamental control principle of the brain suggesting that inhibitory control is essential for high motor efficiency and that a breakdown of inhibition results in more and more effortful movements. Although we only investigated healthy young adults, our study provides the testable hypothesis that an abnormal release of inhibition within M1 might contribute to pathological fatigability and/or bradykinetic movements.

## Materials and methods

### General behavioural paradigm and analysis

All participants were healthy adults, free of medication, had no history of neurological or psychiatric disease and were naïve to the purpose of the experiment. All experimental protocols were approved by the research ethics committee of the canton of Zurich (KEK-ZH 2014–0242, KEK-ZH 2014-0562 or KEK-ZH 2015–0537) and participants gave informed consent to the study.

In the first two experiments we used multiple behavioural conditions. In experiment 1, we compared tapping durations of 10 s, 30 s and 50 s at maximal speed and in experiment 2, we compared tapping at maximal speed with paced tapping either at maximal or submaximal speeds. In all

subsequent experiments (3-9) we used two behavioural conditions manipulating tapping duration: (i) tapping at maximal speed had to be maintained for at least 30 s causing substantial motor slowing (motor slowing condition); and (ii) tapping at maximal speed had to be maintained for only 10 s causing only minor motor slowing (control condition). Each condition was followed by a break of at least 30 s (except for experiment three where the break length varied from 5 to 30 s to perturb recovery, see below). In all experiments (except experiment 2), the order of the experimental conditions was pseudo-randomised. The individual experiments varied in terms of effectors involved (i.e. different repetitive finger sequences, foot or eye movements) and measurement methods (fMRI, EEG, TMS, EMG). For comparison across different effector types and tasks, we analysed the movement speed in number of cycles per bin of 10 s (if not stated otherwise). A cycle was defined as the time from movement of the first effector through the whole movement sequence, back to the first effector. For each participant the average movement speed was calculated for the slowing and control condition, then the movement speed of the slowing condition was normalised to the control condition. Statistical analyses were performed using linear mixed effects models (LMEM) in SPSS 24 (IBM, New York, USA). Typically, the LMEM contains the fixed factor *time* (first-last bin) and the random factor *participant*. Presence of motor slowing was defined as a significant decrease of movement speed from the first to the last bin of the normalised slowing condition (i.e. main effect of time).

## Additional information for specific experiments

### Experiment 1: Time-course of motor slowing

23 healthy volunteers participated in this experiment (12 female, age 23.8 ± 3.0, right handed).

#### Behavioural paradigm and data analysis

In this experiment we characterised the time course of motor slowing by comparing movements at maximal speed of different lengths. Participants were instructed to repetitively tap as fast as possible with the index and middle finger of their dominant hand for 10 s, 30 s or 50 s followed by a break of 50 s for recovery. Participants performed 12 trials per condition (36 trials in total). The order of conditions was pseudo-randomised. After 18 trials there was a 5 min break to reduce potential muscular fatigue. For data analysis, we calculated the movement speed between taps of the same finger. The data within each 1 s bin was then averaged to get a time course of the movement speed over different tapping durations. The data were then normalised to the first 1 s bin of the 10 s condition. Qualitative analysis revealed that participants slowed down immediately with movement onset and reach a 'plateau' after approximately 30 s. For subsequent experiments (except experiment 2) we therefore decided to compare 10 s of tapping (less slowing, control condition) to 30 s of tapping (slowing condition).

### Experiment 2: Speed-dependence of motor slowing

12 healthy volunteers participated in this experiment (four female, age 29.5 ± 6.5 years, one left-handed).

#### Behavioural paradigm and data analysis

In this experiment we characterised whether motor slowing is dependent on the initial movement speed. The experiment was conducted in two parts. In the first part, participants performed 30 s of finger tapping with the index finger of their dominant hand at maximal speed (five trials), which again lead to characteristic motor slowing. In the second part of the experiment participants were then paced at either their initial speed (i.e. the speed in the initial 10 s; fast pacing), their final speed (i.e. the speed in the last 10 s; slow pacing) or at 90% of their final speed (ultraslow pacing). Again, participants performed five trials per condition and the order of conditions was pseudo-randomised. The pacing was provided in form of an online feedback. Participants were instructed to keep a blue bar, which represented a 1 s sliding average of their movement frequency, within a white box, which represented the target frequency ± 0.2 Hz. If participants were moving at the target speed the bar turned green, otherwise it remained blue. For statistical analysis we calculated the movement speed between finger taps in 10 s bins. The data were then submitted to a linear mixed effects model

involving the fixed factors *time* (1st-3rd bin) and *condition* (slowing, fast pacing, slow pacing and ultraslow pacing) and the random factor *participant*.

## Experiment 3: Repetitive foot movements
12 healthy volunteers participated in this experiment (six female, age 27.5 ± 8.43 years, right handed).

### Behavioural paradigm and data analysis
In this experiment, we assessed the presence of motor slowing during foot tapping. Therefore, we compared a slowing condition (30 s of repetitive alternating left-right foot tapping) to a control condition (10 s of the same tapping task performed in the slowing condition). After each tapping condition, there was a break of 40 s for recovery. Participants conducted 10 trials per condition. We calculated the movement speed between each alternating foot tap, averaged these data within three 10 s bins for each experimental condition and submitting it to a LMEM as described above.

A custom-built tapping device was equipped with a force sensor (FSR Model 406, Interlink Electronics Inc, California, USA) to record single taps. The foot tapping was characterised by lifting the heel from the tapping sensor, which was fixated on a ground plate. The forefoot was always in contact with the ground. The main muscles involved in this movement are *M. gastrocnemius* and *M. soleus*. We excluded competitive athletes (i.e. more than 10 hr of training per week) from the study.

## Experiment 4: Repetitive eye movements
12 healthy volunteers participated in this experiment (10 female, age 28.8 ± 10.31 years, right handed).

### Behavioural paradigm
In this experiment we assessed the presence of motor slowing during eye movements. We compared a slowing condition (30 s of repetitive eye movements) to a control condition (10 s of the same tapping task performed in the slowing condition). After each condition, there was a break of 40 s for recovery. During rest, participants were allowed to close their eyes for relaxation and wetting. An auditory preparation cue indicated the start of a new trial. Participants conducted 10 trials per condition. The experimental set-up consisted of an eye-tracker with the corresponding monitor (Tobii TX300 Eye Tracker, Tobii Technology, Stockholm, Sweden; sampling rate 120 Hz) and a custom-made chin rest. Participants were instructed to move their eyes as fast as possible between a left and right target on the screen and blink as little as possible. The target was a red fixation cross on a grey square (size 7 × 9 cm; angular size 2.67 × 3.44°). The target disappeared as soon as an eye movement that reached the target area was detected (margin 1 cm). A short familiarisation session was conducted before the experiment started. The main muscles involved in this type of eye movement are the lateral and medial rectus eye muscles. We excluded participants with eye conditions and/or glasses from the study.

### Data analysis
The point of gaze was calculated by averaging the position of the left and right eye. Based on that, we determined the time needed to shift the gaze from the first to the second target and back, corresponding to one movement cycle. From these data, we determined the movement speed, which was averaged within three 10 s bins. Data from the motor slowing condition (30 s tapping) were normalised to the control condition (10 s tapping) and submitted to a LMEM that included the fixed factor *time* (1 st – 3rd bin) and the random factor *participant*.

## Experiment 5: Repetitive finger movements and characterising the recovery period
17 volunteers participated in this experiment (13 female, age 23.9 ± 3.41 years, all right handed).

### Behavioural paradigm
Here we assessed motor slowing and the time course of its recovery during the subsequent break. Participants performed slowing conditions (30 s of repetitive alternating tapping of index and middle

finger) and control conditions (10 s of the same tapping task performed in the slowing condition). The crucial experimental manipulation is that we varied the length of the break after tapping in 5 s steps (i.e. 5, 10, 15, 20, 25, 30 s) and investigated how break length influences motor slowing recovery (*Figure 3*). The experiment was conducted in four experimental blocks: 2 blocks required 10 s tapping episodes interleaved with breaks and 2 blocks required 30 s tapping episodes interleaved with breaks. The order of these blocks was randomised across participants. Each block consisted of 31 tapping trials separated by 30 breaks. Within each block, the break length pseudo-randomly varied and 10 trials per break condition (i.e. 5, 10, 15, 20, 25, or 30 s length) were performed.

## Data analysis
First, we characterised tapping speed during the tapping episode by calculating the time period between two taps of the same finger. From this data, movement speed was calculated, averaged across three 10 s bins and the data of the motor slowing condition (30 s tapping) were normalised to the control condition (10 s tapping). Next, data were grouped according to the length of the preceding or following break and subjected to a LMEM including the fixed factors *time* (1st-3rd bin), *break length* (5, 10, 15, 20, 25, 30) and the random factor *participant*.

## Recovery
Next, we calculated a recovery index by subtracting the average movement speed of the last 10 s before a break from the average movement speed of the first 10 s after a break. A higher recovery index indicates greater tapping speed recovery during the break. The recovery index was then submitted to a LMEM with the fixed factors *condition* (slowing vs. control) and *break length* (5, 10, 15, 20, 25, 30) to statistically assess the difference between the slowing and control conditions. Next, we estimated (i) the slope of motor slowing (i.e. decrease in movement speed) via linear regression from the movement speed across the three time bins (collapsed across break-length) and (ii) the slope of the recovery index (i.e. recovery speed) via a linear regression for each individual. The relationship between movement speed and recovery speed was then assessed using Pearson's r.

## Experiment 6: fMRI experiment
### Participants
In the first neuroimaging experiment, we applied fMRI while participants executed slowing versus control tapping conditions. 25 participants participated in the experiment (13 female, mean age: 23.6 ± 3.4, right handed).

### Behavioural task and analysis
The experiment consisted of intervals of either slowing (30 s) or control (10 s) tapping with the index and middle finger, followed by a 30 s break. Before each condition participants were shown a visual get ready signal (randomly jittered between 2–3 s). The conditions were blocked within each fMRI run, that is four trials of 30 s tapping were followed by four trials of 10 s tapping (or vice versa). Participants performed two runs, each consisting of 2 blocks, leading to 16 trials per condition. The order of conditions was counterbalanced across runs and the starting condition (i.e. whether the first run started with 10 or 30 s tapping) was counterbalanced across participants. Additionally, after each block there was an implicit baseline condition of 20 s added (i.e. after a 30 s break when recovery was completed). Behavioural data were analysed as described for Experiment three and the normalised movement speed was subjected to a LMEM with the fixed factor *time* and the random factor *participant*. Motor slowing was defined as a significant main effect of *time*.

### fMRI acquisition and preprocessing
FMRI scans were acquired with a Philips Ingenia 3T whole body scanner. Prior to the experiment, high resolution T1-weighted anatomical scans were acquired and used for image registration and normalisation (voxel size = 1 mm3, 160 sagittal slices, matrix size = 240×240, TR/TE = 8.3/3.9 ms). During the behavioural paradigm 360 echo planar images (EPIs) were acquired (voxel size = 2.75×2.75×3.3 mm, matrix size = 128×128, TR/TE = 2500/35 ms, flip angle = 82, 40 slices acquired in interleaved order for full brain coverage). Preprocessing was performed using SPM12 (Wellcome Trust) with default parameters and consisted of the following steps: First, we spatially realigned all EPIs to the average EPI to correct for potential head movements. Second, the

anatomical image was segmented which revealed three probability maps for grey matter, white matter and cerebral spinal fluid and a forward transformation to normalise the individual T1 image to MNI space. The probability maps were used to generate a skull stripped structural image (see default preprocessing pipeline of SPM 12). Third, EPIs were coregistered to the T1 using normalised mutual information and normalised to MNI space by applying the forward transformation resulting from the segmentation step. These normalised images (2×2×2 mm voxels) were spatially smoothed with an isotropic Gaussian kernel of 8 mm full-width at half-maximum.

## FMRI data analysis

All fMRI analyses were performed using SPM12. We first performed a parametric analysis. The first-level model of each participant consisted of a fixed-effects general linear model (GLM). The GLM design matrix included two regressors of interest: The first regressor reflected the tapping periods; the second regressor consisted of a parametric modulator reflecting a linear increase of motor slowing over time. Importantly, the movement speed was orthogonalised with respect to tapping. Note that the slowing and control conditions were modelled together (i.e. slowing consisted of a linear increase in 6 bins of 5 s, and control consisted of a linear increase in 2 bins of 5 s). We also modelled the get-ready periods (see behavioural task above) to regress out visual activation. All conditions were then convolved with a canonical hemodynamic response function (HRF) to account for the hemodynamic delay. Six head movement parameters (translation and rotation along the x, y and z-axis) estimated during realignment were added as regressors of no interest. The two regressors of interest were contrasted against the implicit baseline and then subjected to a second-level random-effects analysis across participants. The second level analysis was a single t-test contrasting the tapping and the slowing against zero. P-values smaller than 0.05 family-wise error (FWE) corrected for multiple comparisons were considered statistically significant.

Additionally, we also analysed the data via a conventional block design. The first-level model for each participant consisted of a fixed-effects GLM which included 10 regressors of interest. For the slowing conditions (30 s tapping followed by 30 s rest), we modelled the first, second and third 10 s bin during tapping and the first, second and third 10 s bin of the subsequent break. For the control condition (10 s tapping followed by 30 s break), we modelled one 10 s-block for the tapping period and the first, second and third 10 s bins of the subsequent break. We also modelled the get-ready periods (see behavioural task above), to regress out visual activation. All conditions were then convolved with a canonical hemodynamic response function (HRF) to account for the hemodynamic delay. Again, head movement parameters were added as regressors of no interest. The ten regressors of interest were contrasted against the implicit baseline and entered into a second-level random-effects analysis across participants.

The second-level model was a flexible (fractional) factorial design consisting of the factors condition (2-levels) and time (6-levels for the slowing condition, that is 3 levels of tapping, three levels during the break and 4-levels for the control condition, that is 1 level of tapping and 3 levels of break). The second-level analyses focused on identifying brain regions that were active during the motor task (i.e. average activation of tapping), and that changed activity over the course of motor slowing (i.e. contrasting the first and last 10 s of long tapping). Additionally, we identified areas that exhibited a differential recovery during the break after the slowing vs. control condition (i.e. showed a significant *condition × time* interaction during the break).

Finally, we contrasted the first 10 s of tapping during the slowing condition with the 10 s of control condition tapping to exclude systematic differences which might have been caused by pacing or other strategies. A p-value smaller than 0.05 family-wise error (FWE) corrected for multiple comparisons was considered statistically significant. Localisation of functional clusters was aided by the anatomy toolbox (*Eickhoff et al., 2005*).

## Experiment 7: EEG experiment

### Participants

Here we combined the behavioural paradigm with electroencephalography (EEG) to assess changes in the alpha (mu-rhythm, 8–12 Hz), beta (14–30 Hz) and gamma bands (30–40 Hz), three intrinsic rhythms of sensorimotor cortex. 17 participants participated in the EEG experiment (10 female, mean age: 25.6 ± 4.1, right handed).

## Behavioural paradigm and data analysis

Similar to experiment four participants performed slowing (30 s tapping) and control trials (10 s tapping) followed by a break of 30 s. The tapping was performed with four fingers (index, middle, ring and little finger) of the left hand and participants were instructed to repetitively perform a pre-trained sequence (4-1-3-2-4; 1 = little, 2 = ring, 3 = middle and 4 = index finger) as quickly and accurately as possible. Twelve trials were performed per condition, leading to 24 trials in total. The order of trials was pseudo-randomised. Tapping speed during the tapping episode was characterised by calculating the time period necessary to complete a sequence (i.e. time from first tap of a sequence to the first tap of the next sequence). From this data, movement speed was calculated, averaged across three 10 s bins and the data of the motor slowing condition (30 s tapping) were normalised to the control condition (10 s tapping). The normalised movement speed was then subjected to a LMEM with the fixed factor time and the random factor participant.

## EEG acquisition

EEG was acquired during the whole experiment using a 128-channel HydroCel Geodesic Sensor Net (GSN) with Ag/AgCl electrodes provided by Electrical Geodesics (EGI, Eugene Oregon, USA). This system uses the vertex (Cz) electrode as a physical reference. EEG recordings, electrooculograms for horizontal and vertical eye movements, and an electromyogram for the muscular noise associated with swallowing were recorded in parallel with a sampling frequency of 1000 Hz. During acquisition participants sat in a dimly lit room in front of a computer screen and performed a finger tapping task at maximum voluntary speed using the paradigm described above.

## EEG preprocessing

Since EEG measurements during ongoing tapping are susceptible to non-neural movement artefacts, all analyses were performed for data during the break, that is when participants were resting. The analysis of the EEG data was performed offline using EEGLAB (*Delorme and Makeig, 2004*). EEG signals during the break were bandpass filtered off-line (3–40 Hz) and processed using independent component analysis (ICA) for the removal of ocular and muscular artefacts. After ICA decomposition the artefact ICs were automatically detected by correlating their power time-courses with the power time courses of the electric reference signals (horizontal and vertical electrooculogram and electromyogram). The data were down-sampled to 200 Hz and re-referenced to the common average (*Liu et al., 2015*) to remove the bias towards the physical reference site.

## EEG source localisation

After preprocessing source localisation of the EEG data was performed to extract the EEG signals from the three à priori defined regions of interest which showed increased activity with increasing motor slowing (i.e. SM1, SMA and PMd). A forward head model was built with the finite element method (FEM) using a 12-tissue head template and the standard electrode positions for a 128-channel EGI cap. The head template was obtained from the IT'IS foundation of ETH Zurich (*Iacono et al., 2015*) and included 12-tissue classes (skin, eyes, muscle, fat, spongy bone, compact bone, cortical grey matter, cerebellar grey matter, cortical white matter, cerebellar white matter, cerebrospinal fluid and brain stem). Specific conductivity values were associated with each tissue class (i.e. skin 0.4348 S/m, compact bone 0.0063 S/m, spongy bone 0.0400 S/m, CSF 1.5385 S/m, cortical grey matter 0.3333 S/m, cerebellar grey matter 0.2564 S/m, cortical white matter 0.1429 S/m cerebellar white matter 0.1099 S/m, brainstem 0.1538 S/m, eyes 0.5000 S/m, muscle 0.1000 S/m, fat 0.0400 S/m; *Haueisen et al., 1997*). The dipoles corresponding to brain sources were placed on a regular 6 mm grid spanning cortical and cerebellar grey matter. After the head model template was established, the brain activity in each dipole source was estimated by the exact low-resolution brain electromagnetic tomography (eLORETA; *Pascual-Marqui et al., 2011*) for each participant.

## EEG data analysis

From source-localised EEG data, we extracted the first principle component from three regions of interest derived from the group peak-activation reflecting motor slowing in the fMRI experiment (SMA, MNI −6 -8 50; left PMd, MNI −28 -16 70 and left SM1, MNI −34 -20 55).

The data were then analysed for 30 s breaks after tapping. For each participant the signal was filtered to the alpha (8–14 Hz), beta (15–30 Hz) and gamma (30–40 Hz) band, rhythms which are

classically associated with sensorimotor function. Then the data were rectified and smoothed with a sliding average filter (length 1 s, no overlap) to get an estimate of the amplitude over time. For statistical analysis the recovery time course was binned into three 10 s epochs (0–10 s, 10–20 s, 20–30 s), and the mean amplitude per block was determined. The amplitude data of each frequency was then subjected to a LMEM with the fixed repeated factors *condition* (slowing vs. control tapping) and *time* (0–10 s, 10–20 s, 20–30 s) and the random factor *participant*.

### Experiment 8: SICI experiment

#### Participants

Here we combined the behavioural paradigm with TMS to probe GABAergic inhibition during the recovery period. 13 participants participated in the experiment (four female, mean age: 24.8 ± 2.5, all right handed) over two sessions. None of the participants reported contraindications to TMS.

#### Behavioural paradigm

The behavioural paradigm consisted of sequential tapping with four fingers of the right hand. The pre-trained sequence was 0-1-3-2-0 (0 = thumb, 1 = index, 2 = middle and 3 = ring finger). Again, we compared slowing (40 s of tapping) and control (10 s of tapping) conditions, each followed by a 40 s break. The two conditions were split into two sessions on two consecutive days (counter-balanced across participants) with 12 trials each. Behavioural data were analysed following the same procedures as for Experiment 5.

#### TMS protocol

We assessed GABAergic inhibition in the primary motor cortex by using a short-interval intracortical inhibition TMS protocol (*Kujirai et al., 1993*). In short, SICI is measured with a paired-pulse (DP) TMS protocol where a subthreshold conditioning stimulus (CS) is applied shortly (2 ms) before a suprathreshold test stimulus (TS). Typically, the amplitude of motor evoked potentials (MEP) is attenuated when the CS+TS condition is compared to the TS condition only. This reduction has been linked to the activity of GABA$_A$-ergic inhibitory circuits (*Werhahn et al., 1999*; *Ziemann et al., 1996*). Since SICI measurements are most reliable during rest, it was assessed during the breaks after tapping, that is during the recovery period.

TMS was performed with a figure-of-eight coil (70 mm) connected to two Magstim Bi-Stim2 stimulators (Magstim, Withland, UK) and electromyography (EMG) was measured from the first dorsal interosseus (FDI) and abductor pollicis brevis (APB) muscles. First, the hotspot for the FDI muscle was determined, that is the location with the largest and most consistent MEP. Neuronavigation (Brainsight, Rogue Research, Montreal, Canada) was used to ensure a constant coil position during the experiment. The coil was positioned over the left hemisphere and held tangentially with the handle pointing backward and laterally at 45° away from the mid-sagittal line. Rest motor threshold (RMT) was then defined as the lowest stimulus intensity eliciting MEPs, which were larger then 50µV in at least five out of ten trials. The TS was set to 130% of RMT. For SICI the interstimulus interval was set to 2 ms and the intensity of the CS was chosen such that it reduced MEP amplitude relative to the TS only condition by approximately 50%. Once the stimulation parameters were set, a Pre measurement consisting of 18 CS+TS and 18 TS (jittered inter stimulus interval $\geq$ 4 s) was obtained before the behavioural experiment. During the tapping experiment, TMS was applied during the 40 s break so that 3 CS+TS and 3 TS were measured between second 3.5 and 33.5 in pseudo-randomised order. This resulted in 24 CS+TS and 24 TS for each tapping condition and time point. After the tapping experiment, a post measurement was performed which again consisted of 18 CS+TS and 18 TS.

#### TMS data analysis

All analyses focused on FDI muscle effects since it was used to define the hotspot and RMT. Background EMG was quantified by taking the root mean square of the EMG signal between 10 and 110 ms before the first TMS pulse was delivered. Trials with background EMG above 0.1 mV were removed from further analyses. For the remaining trials, mean and standard deviation of the background EMG was calculated for each participant, and for the TS and CS+TS trials separately. Trials with a background EMG > mean ± 2.5 standard deviations were also excluded from further analysis. During all TMS measurements, MEP size was determined as the peak-to-peak amplitude. MEPs were

considered outliers and excluded from the analysis if they were greater than Q3+1.5× (Q3-Q1) or less than Q1-1.5× (Q3-Q1), where Q1 and Q3 are equal to the first and third quartiles, respectively (*Zhang et al., 2014*). Based on these criteria 83% of the trials were included in further analyses. SICI was then calculated according to the following formula: Inhibition = (1-(CS+TS/TS)), that is low values indicate low inhibition while large values indicate high inhibition.

Inhibition was averaged separately for the Pre and Post measurements, as well as for three time intervals during the break (3.5–22 s, 7.5–27.5 s, 12.5–33.5 s) either following the slowing condition (i.e. during pronounced recovery) or the control condition (i.e. during minor recovery). The data were then subjected to a LMEM with the fixed factors condition (slowing vs. control) and time (Pre, 3.5–22 s, 7.5–27.5 s, 12.5–33.5 s, Post), and the random factor participant.

## Experiment 9: Surround inhibition experiment
### Participants
Twenty-six adults (14 females, mean age and standard deviation: 24 ± 3 years, range: 18–32 years) participated in the experiment after providing written informed consent. All participants were right handed (mean laterality index and standard deviation 93 ± 12%, range: 60–100%). None of the participants reported contraindications to TMS. Seven participants had to be excluded because they did not show a clear surround inhibition effect during the Pre measurement (see below) leading to a final sample size of 19 participants.

### General setup
Participants were comfortably seated in front of a desk. Their right forearm rested on the desk in a neutral position with a slight shoulder abduction and about 60° elbow flexion. The palm and the forearm of the participants were strapped to a custom-made wooden structure which only allowed movement of the thumb. There was a computer monitor placed about 70 cm in front of the participant.

Surface electromyography (BagnoliTM, Delsys, USA) was recorded from the right first dorsal interosseus (FDI), abductor pollicis brevis (APB), abductor digiti minimi (ADM) and extensor pollicis longus (EPL). EMG data were sampled at 5000 Hz (CED Power 1401, Cambridge Electronic Design, UK), amplified, band pass filtered (5–1000 Hz), and stored on a PC for off-line analysis.

### Behavioural paradigm
The motor slowing (30 s tapping followed by 60 s break) and control conditions (10 s tapping followed by 60 s break) were tested on two different days (order counterbalanced across participants). During each testing session, participants were asked to tap with their right thumb as fast as possible. Tapping was measured with a custom made device incorporating a vertical force sensor. Participants had their hand placed upright and performed horizontal taps against the vertical sensor. In the first session, participants were introduced to all measurements and performed a familiarisation block. Each session consisted of 20 trials in total. Behavioural data were analysed by calculating time between taps. From this data, movement speed was calculated, averaged across three 10 s bins for each condition and subjected to a LMEM.

### Coactivation analysis
Coactivation of EPL and APB, the two main effectors during thumb tapping, was determined by calculating the overlap in EMG signals. EMG signals were recorded at 1000 Hz during tapping and analysed offline. EMG signals were high-pass filtered at 20 Hz and rectified, then a 10 Hz low-pass filter was used on the rectified signal to extract the envelope. The amount of coactivation (coactivation index) was defined as the joint area under the curve of the two signals, normalised to the area under the curve of the agonist (i.e. APB; *Frost et al., 1997*). The coactivation index was calculated and averaged across trials in bins of 10 s and the data of the slowing condition (30 s) were normalised to the data of the control condition (10 s) and subjected to a LMEM.

### TMS preparation
TMS was performed using the same principle approach as described in experiment 6. In each testing session a hotspot was determined that consistently evoked MEPs in both the ADM and FDI muscles. Rest motor threshold (RMT) of the ADM was determined to the nearest 1% of maximum stimulator

output. ADM was chosen as the target muscle because pilot experiments had revealed that the resulting RMT was sufficiently high to evoke consistent MEPs in both muscles of interest. During the testing sessions stimulation intensity was set to 140% of the RMT.

## Determining surround inhibition

The main aim of experiment seven was to determine surround inhibition after slowing vs. control tapping. Surround inhibition in humans can be measured by comparing the excitability between movement execution and rest of a non-involved muscle, surrounding the actual movement effector. Here, we used the EPL as movement effector and measured the excitability of FDI either during movement execution of EPL or at rest. To that end, participants were instructed to perform a brief tap after a beep tone. TMS was triggered by the EMG activity of EPL (self-triggered TMS by the thumb extensor). The trigger level of the EMG activity was set at 100 µV peak-to-peak EMG amplitude (this value is chosen to avoid triggering while resting; *Sohn and Hallett, 2004*). Half of the TMS pulses were delivered 3 ms after the trigger ($TMS_{Move}$) to probe surround inhibition while the other half were delivered 2 s after the trigger ($TMS_{Con}$), which served as a control condition since participants were again at rest. The order of TMS pulses were semi-randomised. 'Beep' tones were given at random intervals between 7 and 9 s. Participants were asked to tap horizontally on the sensor board briefly after each beep tone with a self-paced delay. All participants were instructed that it was not necessary to react as fast as possible.

These measurements were performed during the experiment (Pre) and immediately after the tapping intervention (Post), as well as during the break after each tapping trial. In both Pre and Post TMS measurements, 15 $TMS_{Move}$ and 15 $TMS_{Con}$ pulses were applied. During each 60 s break following either motor slowing or control tapping, 3 $TMS_{Move}$ and $TMS_{Con}$ pulses were applied. The order of TMS pulses was semi-randomised. The first tone was played 5 s after the last tap and subsequent tones were played randomly every 7–9 s. Surround inhibition was determined by taking the quotient of $TMS_{Move}$ divided by $TMS_{Con}$.

## Data analyses TMS

For all TMS measurements, MEP size was determined by peak-to-peak amplitude. MEPs and background EMG values were tested for outliers as described in Experiment six and these trials were excluded from the analysis. 96% of trials (APB = 95%, FDI = 96%, EPL = 97%, ADM = 96%) were included in the analysis. Surround Inhibition was then calculated for the FDI muscle by dividing the MEPs from $TMS_{Move}$ by the MEPs from $TMS_{Con}$. A value smaller than one reflects inhibition of FDI relative to EPL/APB, whereas a value larger than one reflects facilitation of FDI relative to EPL/APB. Surround Inhibition was calculated separately for Pre and Post, as well as for the three time points during the break (5–14 s, 19–32 s, 33–50 s). Since we were interested in the change of surround inhibition during slowing relative to the control condition, we normalised the data during the break (5–14 s, 19–32 s, 33–50 s) to the individual Pre measurement. The data were then subjected to a LMEM with the fixed factors *condition* (slowing vs. control) and *time* (5–14 s, 19–32 s, 33–50 s), and the random factor *participant*.

## Acknowledgements

The authors would like to thank Xue Zhang, Marta Stepien, Marionna Münger, Alex Hess, Andrea Bosshard and David Tanner for data collection during the behavioural and/or TMS experiments, Dan Woolley for proof-reading the manuscript and technical assistance while performing the experiments, Charles Lambelet for building a supportive device for the surround inhibition experiment, Dante Mantini for providing code for source localisation of the EEG data and Sarah Meissner for critical comments on the manuscript. This work was supported by the Swiss National Science Foundation (SNSF 320030_175616) and the Singapore-ETH Centre.

## Additional information

### Funding

| Funder | Grant reference number | Author |
| --- | --- | --- |
| Swiss National Science Foundation | SNSF 320030_175616 | Marc Bächinger<br>Rea Lehner<br>Felix Thomas<br>Nicole Wenderoth |
| Singapore-ETH Centre | | Rea Lehner<br>Nicole Wenderoth |

The funders had no role in study design, data collection and interpretation, or the decision to submit the work for publication.

### Author contributions

Marc Bächinger, Rea Lehner, Conceptualization, Resources, Data curation, Software, Formal analysis, Investigation, Visualization, Methodology, Writing—original draft, Project administration, Writing—review and editing; Felix Thomas, Conceptualization, Data curation, Formal analysis, Investigation, Writing—review and editing; Samira Hanimann, Data curation, Formal analysis, Investigation, Writing—review and editing; Joshua Balsters, Conceptualization, Formal analysis, Supervision, Writing—original draft, Writing—review and editing; Nicole Wenderoth, Conceptualization, Formal analysis, Supervision, Funding acquisition, Methodology, Writing—original draft, Project administration, Writing—review and editing

### Author ORCIDs

Marc Bächinger (iD) https://orcid.org/0000-0002-3726-542X
Rea Lehner (iD) https://orcid.org/0000-0002-6497-2875
Joshua Balsters (iD) http://orcid.org/0000-0001-9856-6990
Nicole Wenderoth (iD) https://orcid.org/0000-0002-3246-9386

### Ethics

Human subjects: All experimental protocols were approved by the research ethics committee of the canton of Zurich (KEK-ZH 2014-0242, KEK-ZH 2014-0562, KEK-ZH 2015-0537) and participants gave informed consent to the study.

### Decision letter and Author response

Decision letter https://doi.org/10.7554/eLife.46750.040
Author response https://doi.org/10.7554/eLife.46750.041

## Additional files

### Supplementary files

• Supplementary file 1. Table showing individual movement speeds for each condition of experiment 2, including the different target speeds of each participant.
DOI: https://doi.org/10.7554/eLife.46750.030

• Supplementary file 2. Table showing peak fMRI activations during tapping.
DOI: https://doi.org/10.7554/eLife.46750.031

• Supplementary file 3. Table showing rest motor threshold, conditioning stimulus and test stimulus intensities of SICI measures as % maximum stimulator output (%MSO), showing that stimulus intensities were comparable across sessions (all values mean ± standard deviation).
DOI: https://doi.org/10.7554/eLife.46750.032

• Supplementary file 4. Table showing background EMG (mV) of SICI measures during the break. Note that background EMG was very small and that potential differences were <0.001 mV and well below the observed standard deviation. Statistical tests revealed only non-significant effects of

*condition* (F(1,12)=0.696, p=0.42), or *time* (F(2,12)=3.137, p=0.08). Most importantly, there was no significant *condition x time* (F(2,12) = 0.203, p=0.819) interaction for background EMG. All values mean ± std.

DOI: https://doi.org/10.7554/eLife.46750.033

• Supplementary file 5. Table showing average motor evoked potentials of the control stimulus in mV for Pre and Post measurements, as well as during the break after 30 s and 10 s tapping showing that the control stimulus did not change during the surround inhibition experiment. Further, the rest motor threshold (RMT, in percentage of maximum stimulator output) was comparable across both sessions.

DOI: https://doi.org/10.7554/eLife.46750.034

• Supplementary file 6. Table showing background EMG of surround inhibition measures during the break. Note that background EMG was very small and that potential differences were <0.002 mV and well below the observed standard deviation. Statistical tests revealed only non-significant effects of *condition* (F(1,18)=1.728, p=0.205) or *time* (F(2,18)=0.701, p=0.509). Most importantly, there was no significant *condition x time* (F(2,18) = 0.735, p=0.493) interaction for background EMG. All values mean ± std.

DOI: https://doi.org/10.7554/eLife.46750.035

• Supplementary file 7. Data from previous studies used for sample size calculations as described in the transparent reporting form.

DOI: https://doi.org/10.7554/eLife.46750.036

• Supplementary file 8. Summary table of the statistical tests performed in each experiment as described in the transparent reporting form.

DOI: https://doi.org/10.7554/eLife.46750.037

• Transparent reporting form

DOI: https://doi.org/10.7554/eLife.46750.038

## Data availability

All data generated or analysed during this study are included in the manuscript and supporting files. Source data files have been provided for all the figures. Statistical maps of the main fMRI results have been provided.

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
