## [Decision Letter]

Thank you for submitting your article "Motor fatigability as evoked by repetitive movements results from a gradual breakdown of surround inhibition" for consideration by *eLife*. Your article has been reviewed by three peer reviewers, including Nicole Swann as the Reviewing Editor and Reviewer #1, and the evaluation has been overseen by Richard Ivry as the Senior Editor. The following individuals involved in review of your submission have agreed to reveal their identity: Sven Bestmann (Reviewer #2); Hayley MacDonald (Reviewer #3).

The reviewers have discussed the reviews with one another and the Reviewing Editor has drafted this decision to help you prepare a revised submission.

The authors have conducted an impressive set of experiments which demonstrate that repeated, sub-maximal movements are associated with "motor fatigue" as evidenced by motoric slowing (as measured from multiple effectors). This slowing is associated with increased motor network activation demonstrated using fMRI, alpha power changes measured with EEG, release of inhibition measured with SICI (TMS), and a breakdown of surround inhibition.

The reviewers were impressed with the robust behavioral effect (which was demonstrated in the feet, fingers, and eyes), the use of multiple neuroscientific methods to measure this effect, and the logical flow of the manuscript. We found the notion that motor fatigue relates to the breakdown of surround inhibition particularly interesting.

Essential revisions:

1) We were concerned that there is a confound in the task design wherein the trials with motor fatigue also involve movement of longer duration than the control trials. Thus, we are unsure if the authors can really conclude that the behavioral and neural effects relate to fatigue and not some other aspect of movement duration. One way this question could perhaps be addressed would be by collecting more data (even behavioral data) which showed that motor slowing could occur irrespective of movement duration – for instance by comparing "fatigue" and "control" conditions of the same duration and manipulating "fatigue" by emphasizing speed over accuracy (and vice versa). Alternatively, the authors may be able to address his confound with a re-analysis of their existing data – perhaps by examining trials with a 30 second duration that did not exhibit motor slowing (or had less motor slowing) compared to 30 second trials with substantial slowing. (Additionally, trials with a 10 second direction for which some slowing may have occurred could also be examined compared to 10 second trials without slowing). We leave it up to the authors which of these strategies they would like to take on to address this issue. Of course, they are welcome to take an alternative approach to address this confound.

2) For experiments 6 and 7, we have lingering concerns that differences in RMS EMG could have driven the MEP effects. We noted that trials with RMS EMG values above 0.1mV were excluded, however we would like to see a direct statistical comparison of background RMS EMG between conditions to rule out this potential confound.

[Editors' note: further revisions were requested prior to acceptance, as described below.]

Thank you for resubmitting your work entitled "Motor fatigability as evoked by repetitive movements results from a gradual breakdown of surround inhibition" for further consideration at *eLife*. Your revised article has been favorably evaluated by Richard Ivry (Senior Editor), a Reviewing Editor, and consultation from two reviewers.

The manuscript has been improved but there are some remaining issues that need to be addressed before acceptance, as outlined below:

The reviewers were satisfied with your additional behavioral analyses and feel these add substantially to the publication. However, we felt these analyses, especially the one which showed temporal dependence of the motor speed, shown in Figure 1—figure supplement 1, does change the interpretation of the manuscript fairly substantially. In particular, it suggests that the "fatigue" actually involves an interaction between speed and time. While interesting in either case, we were unsure if this is in line with authors initial framing of "fatigue". Moreover, the additional analyses suggest that the control condition is more of a "less fatigued" versus "no fatigue" condition (which the authors acknowledge). Again, while this does not undermine the importance of the results, we felt that it might warrant more attention and perhaps some reframing in the manuscript. Finally, because the two additional analyses presented in Figure 1—figure supplements 1 and 2 really do impact the interpretation of the behavioral effect, we believe they should be included as main figures (or part of a main figure) and not a supplement.

1) It would be helpful to include behavioral accuracy results for the movement speed analysis presented in Figure 1—figure supplement 2. The reviewers wondered how reliably these speeds could be maintained.

2) *eLife* requires that the Title gives a clear indication of the biological system under investigation. The authors may want to consider clarifying this.

[Editors' note: further revisions were requested prior to acceptance, as described below.]

Thank you for submitting your article "Human motor fatigability as evoked by repetitive movements results from a gradual breakdown of surround inhibition" for consideration by *eLife*.

The Reviewing and Senior Editor have discussed your manuscript and we believe some additional revisions are necessary before acceptance. The Reviewing Editor has drafted this decision to help you prepare a revised submission.

We feel some additional textual edits are necessary to address the first point from our previous decision letter. Specifically, we had indicated that the results from the new control experiments provide important insights into the motor fatigue manipulation (i.e., that it depends on both speed and time). We believe that additional changes to the text in the Results section and/or Discussion section should be made to fully explain and consider these findings. We apologize if this was not clear in the previous decision letter.

---

## [Author Response]

Essential revisions:1) We were concerned that there is a confound in the task design wherein the trials with motor fatigue also involve movement of longer duration than the control trials. Thus, we are unsure if the authors can really conclude that the behavioral and neural effects relate to fatigue and not some other aspect of movement duration. One way this question could perhaps be addressed would be by collecting more data (even behavioral data) which showed that motor slowing could occur irrespective of movement duration – for instance by comparing "fatigue" and "control" conditions of the same duration and manipulating "fatigue" by emphasizing speed over accuracy (and vice versa). Alternatively, the authors may be able to address his confound with a re-analysis of their existing data – perhaps by examining trials with a 30 second duration that did not exhibit motor slowing (or had less motor slowing) compared to 30 second trials with substantial slowing. (Additionally, trials with a 10 second direction for which some slowing may have occurred could also be examined compared to 10 second trials without slowing). We leave it up to the authors which of these strategies they would like to take on to address this issue. Of course, they are welcome to take an alternative approach to address this confound.

The reviewers raise an important point here. To address this concern, we performed two additional behavioral experiments. In short, we first show that motor slowing is time dependent, as would be expected for a process linked to fatigability. Importantly, however, we show in the second experiment that the rate of slowing depends significantly on tapping speed and is not just a function of tapping duration.

In the first behavioral experiment, participants (N=23) were instructed to tap with their index and middle finger (i.e. as in Experiment 3 and 4 of the main manuscript) at maximal rate for 10, 30 or 50s. We analyzed the movement speed in 1s bins. This data shows that motor slowing starts immediately after tapping onset and reaches a plateau after approximately 30s (Figure 1—figure supplement 1). Thus, the motor slowing process is inherently time-dependent indicating that our control condition does not necessarily reflect “no slowing”, but “less slowing”.

In the second behavioral experiment (Figure 1—figure supplement 2), we instructed participants (N=12) to execute repetitive index finger movements at different movement speeds. To do so, we first measured motor slowing and for each participant determined the maximum voluntary speed at the beginning of the trial (first 10s) and the final speed at the end of motor slowing (last 10s). Subsequently, participants had to move for 30s intervals while receiving online feedback of their tapping speed and the required pace. Participants were instructed to keep a blue bar representing the sliding average over 1s of their movement frequency within a white box representing the target speed +/- 0.2 Hz. If participants were moving at the target speed the bar turned green, otherwise it remained blue.

Using this online feedback, they were paced at their (i) initial voluntary speed at the beginning of the slowing trials (fast pacing, green), (ii) their final voluntary speed at the end of the slowing trials (slow pacing, pink), or at an even lower pace corresponding to 90% of the slow pacing speed (ultraslow pacing, red).

This new data shows that slowing is present when participants were paced at their initial voluntary speed (6.18 +/- 5.47 Hz; mean +/- std), but not at their final voluntary speed (5.44 +/- 5.70 Hz) or an ultraslow speed (4.89 +/- 5.13 Hz). This suggests that the observed phenomenon is not dependent on movement duration per se, but rather on whether a high movement speed has to be maintained over a longer period of time. We added the data of this new experiment as Figure 1—figure supplement 2.

2) For experiments 6 and 7, we have lingering concerns that differences in RMS EMG could have driven the MEP effects. We noted that trials with RMS EMG values above 0.1mV were excluded, however we would like to see a direct statistical comparison of background RMS EMG between conditions to rule out this potential confound.

This is a valid concern. We analyzed background EMG as suggested by the reviewers. We did not find a significant difference in background EMG for experiment 6 or 7, see Supplementary file 4 and Supplementary file 6.

[Editors' note: further revisions were requested prior to acceptance, as described below.]

The manuscript has been improved but there are some remaining issues that need to be addressed before acceptance, as outlined below:The reviewers were satisfied with your additional behavioral analyses and feel these add substantially to the publication. However, we felt these analyses, especially the one which showed temporal dependence of the motor speed, shown in Figure 1—figure supplement 1, does change the interpretation of the manuscript fairly substantially. In particular, it suggests that the "fatigue" actually involves an interaction between speed and time. While interesting in either case, we were unsure if this is in line with authors initial framing of "fatigue". Moreover, the additional analyses suggest that the control condition is more of a "less fatigued" versus "no fatigue" condition (which the authors acknowledge). Again, while this does not undermine the importance of the results, we felt that it might warrant more attention and perhaps some reframing in the manuscript. Finally, because the two additional analyses presented in Figure 1—figure supplements 1 and 2 really do impact the interpretation of the behavioral effect, we believe they should be included as main figures (or part of a main figure) and not a supplement.

As suggested by the reviewers, we have added the two figures from the supplement to Figure 1 in the main manuscript. We slightly changed the analysis of the experiment depicted in Figure 1B to make it consistent with the rest of the results. More specifically, we analyzed complete 10s bins (as in all other experiments) while we had omitted the first second of each bin in our previous analysis (to reduce the effect of adjusting to the target speed). Importantly, this does not change the main results of this experiment. We modified the Results section and the Discussion section slightly to integrate these findings into the text and added the individual data as supplementary table.

1) It would be helpful to include behavioral accuracy results for the movement speed analysis presented in Figure 1—figure supplement 2. The reviewers wondered how reliably these speeds could be maintained.

We added an additional analysis comparing the produced tapping speed and the target speed as Figure 1—figure supplement 1.

2) eLife requires that the Title gives a clear indication of the biological system under investigation. The authors may want to consider clarifying this.

We would like to thank the reviewers for pointing that out. We slightly modified the Title to clarify that the investigations were made in humans.

[Editors' note: further revisions were requested prior to acceptance, as described below.]

The Reviewing and Senior Editor have discussed your manuscript and we believe some additional revisions are necessary before acceptance. The Reviewing Editor has drafted this decision to help you prepare a revised submission.We feel some additional textual edits are necessary to address the first point from our previous decision letter. Specifically, we had indicated that the results from the new control experiments provide important insights into the motor fatigue manipulation (i.e., that it depends on both speed and time). We believe that additional changes to the text in the Results section and/or Discussion section should be made to fully explain and consider these findings. We apologize if this was not clear in the previous decision letter.

We made several changes to address this concern:

Introduction:

“We show that motor slowing is a general phenomenon that gradually manifests when high tapping speeds have to be maintained for an extended period of time.”

Results section:

“Together experiment 1 and 2 indicate that motor slowing is a gradual process which depends inherently on the initial speed and the time it is maintained. […] Note that a minor reduction of movement speed is still present during the control condition, albeit significantly less pronounced than in the actual slowing condition.”

Discussion section:

“Note that the motor slowing phenomenon depends on both the initial tapping speed and the duration that tapping must be maintained. […] As such, motor slowing is an interesting paradigm for investigating which central mechanisms might contribute to the more general phenomenon of performance fatigability.”

Materials and methods section:

“In all subsequent experiments (3-9) we used two behavioural conditions manipulating tapping duration: (i) tapping at maximal speed had to be maintained for at least 30s causing substantial motor slowing (motor slowing condition); and (ii) tapping at maximal speed had to be maintained for only 10s causing only minor motor slowing (control condition).”